# Flash drug release from nanoparticles accumulated in the targeted blood vessels facilitates the tumour treatment

Ivan V. Zelepukin [1,2] ✉, Olga Yu. Griaznova [1,2], Konstantin G. Shevchenko[3,4], Andrey V. Ivanov [5], Ekaterina V. Baidyuk[3], Natalia B. Serejnikova [5], Artur B. Volovetskiy[5], Sergey M. Deyev [1,2,5] ✉ & Andrei V. Zvyagin [1,2,5,6] ✉

Tumour microenvironment hinders nanoparticle transport deep into the tissue precluding thorough treatment of solid tumours and metastatic nodes. We introduce an anticancer drug delivery concept termed FlaRE (Flash Release in Endothelium), which represents alternative to the existing approaches based on enhanced permeability and retention effect. This approach relies on enhanced drug-loaded nanocarrier accumulation in vessels of the target tumour or metastasised organ, followed by a rapid release of encapsulated drug within tens of minutes. It leads to a gradient-driven permeation of the drug to the target tissue. This pharmaceutical delivery approach is predicted by theoretical modelling and validated experimentally using rationally designed MIL-101(Fe) metal-organic frameworks. Doxorubicin-loaded MIL-101 nanoparticles get swiftly trapped in the vasculature of the metastasised lungs, disassemble in the blood vessels within 15 minutes and release drug, which rapidly impregnates the organ. A significant improvement of the therapeutic outcome is demonstrated in animal models of early and late-stage B16-F1 melanoma metastases with 11-fold and 4.3-fold decrease of pulmonary melanoma nodes, respectively.

Formulating drug molecules in nanoscale carriers is an attractive approach to improve the therapeutic index of anticancer pharmaceuticals[1,2]. The nanocarriers protect therapeutic cargo from degradation in harsh extracellular environment, mediate transportation to tumour and regulate drug release. It is generally accepted that the passive delivery of nanomedicine to tumour targets occurs via the enhanced permeability and retention (EPR) effect, which relies on nanoparticle (NP) extravasation through endothelial pores or fenestrae and their retention in the tumour interstitium due to the deficient lymphatic drainage[3,4]. Disassembly of nanocarriers into single-digit nano-fragments triggered by the tumour microenvironment has been

previously demonstrated to enhance the diffusion in tumour interstitium and gain access to target cancer cells[5], although this strategy relies on the extravasation of the nanocarriers with associated shortcomings. Other mechanisms of NP extravasation, such as active transcytosis, have also been reported[6,7].

NPs are predominantly accumulated in the perivascular leaky regions of the tumour[8] from where their access to the target cancer cells is hindered by the extracellular matrix and elevated interstitial fluid pressure[9]. Moreover, not all tumour vessels are leaky to NPs due to their structural heterogeneity[10], while the EPR varies across cancer types[8,11]. As a result, only 0.7% of the injected NPs have been reported

[1]Shemyakin-Ovchinnikov Institute of Bioorganic Chemistry of the Russian Academy of Sciences, 117997 Moscow, Russia. [2]National Research Nuclear University MEPhI (Moscow Engineering Physics Institute), 115409 Moscow, Russia. [3]Institute of Cytology of the Russian Academy of Sciences, 194064 Saint Petersburg, Russia. [4]Chumakov Federal Scientific Center for Research and Development of Immunobiological Drugs of the Russian Academy of Sciences, 108819 Moscow, Russia. [5]Sechenov First Moscow State Medical University (Sechenov University), 119991 Moscow, Russia. [6]MQ Photonics Centre, Macquarie University, 2109 Sydney, Australia. ✉e-mail: zelepukin@phystech.edu; deyev@ibch.ru; andrei.zvyagin@mq.edu.au

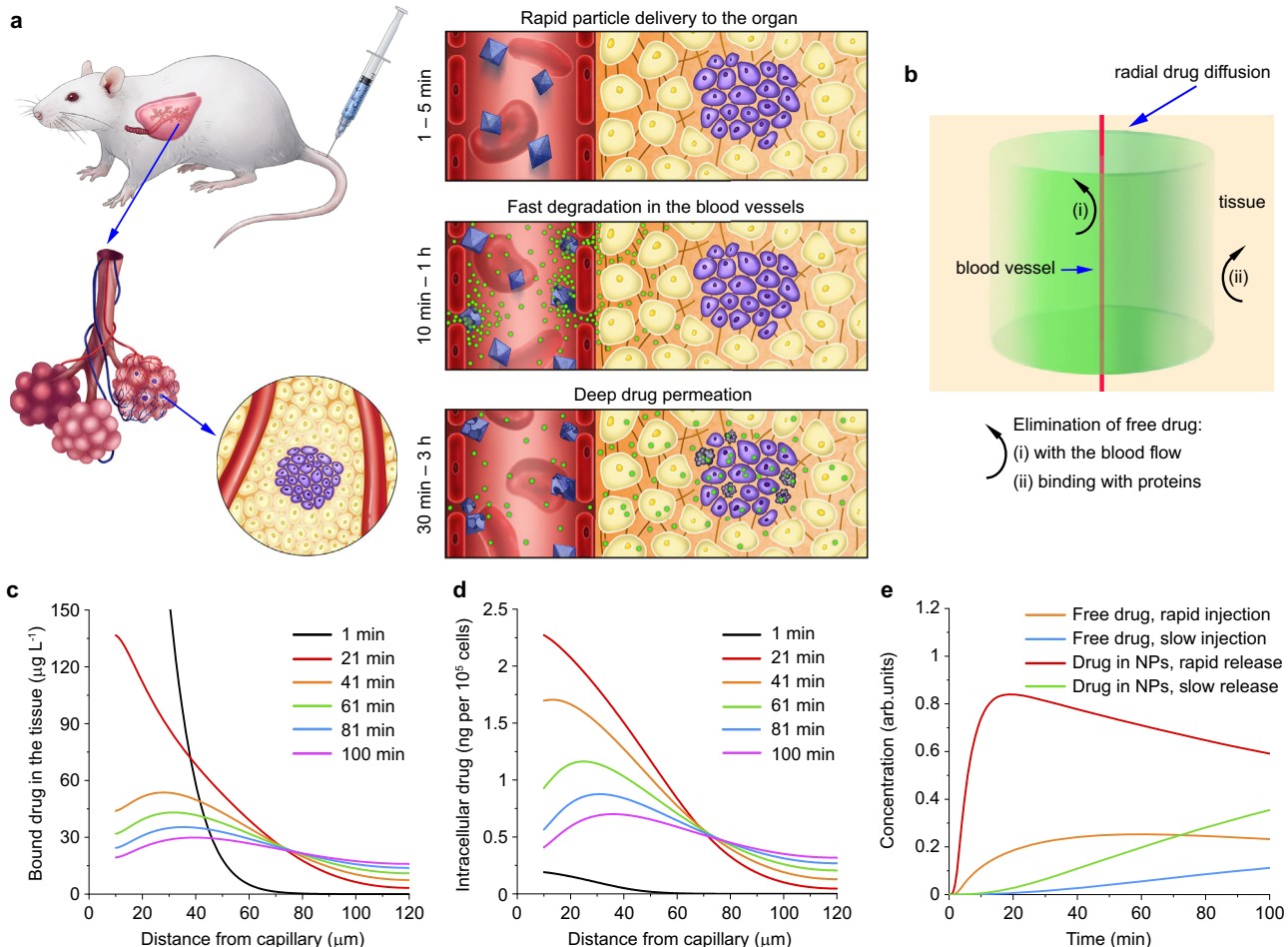

**Fig. 1 | Conceptual blueprint of the FlaRE drug delivery to a metastasised organ. a** Left panel: drug formulation is systemically administered in a mouse and sequestered in capillaries in the metastasised lung tissue (tumour cells shown in purple colour). Right panel shows metal-organic frameworks, rapidly entering capillaries of the lungs and anchored to the endothelium starting to release drugs (shown in green), while undergoing structural transformation. The released drug crosses the vessel walls, reaches the metastatic node by gradient-driven diffusion and damages cancer cells. **b** Schematic diagram of the proposed theoretical drug delivery model. Drugs are considered diffusing from an infinite capillary to uniform interstitial tissue. Details are provided in the Supplementary Note 1. **c**−**e** Theoretical modelling results: plots of the concentration of albumin-bound drug in the interstitium (**c**) and cells (**d**) versus a radial distance from the microcapillary wall, respectively, when the FlaRE drug delivery mode is considered. **e** Plot of the intracellular drug concentration kinetics at 50 μm distance from the microcapillary wall for four cases of the drug delivery. In **c**−**e** kinetics at different time-points are marked with following colours: black (1 min), red (21 min), orange (41 min), green (61 min), blue (81 min), magenta (100 min).

to reach the malignant tissue[12] and <0.0014% were internalised by the cells[13]. Noteworthy, EPR is not a universal mechanism. It has not been observed in clinical samples from solid tumours and is uncommon in small tumours and metastases in animal models[14]. The modest efficiency of ERP-based nanomedicines drives research for alternative drug delivery approaches, which are less dependent on the tumour biology[15].

Here, we introduce a novel non-EPR based concept of drug delivery to primary and metastatic solid tumours, for which we coin the term FlaRE delivery−Flash Release in Endothelium based drug delivery. This approach relies on enhanced drug-loaded nanocarrier accumulation in the microcapillaries of a target tumour or metastasised organ, followed by rapid nanocarrier degradation to release drug to the vessels. The rapidly elevated intracapillary drug concentration drives drug diffusion across the endothelial wall to the surrounding interstitium, spreading deep and reaching cancer cells with ensuing therapeutic effects (Fig. 1a). Note the NP extravasation prerequisite is abrogated in our approach.

The existing concept postulates that it is the gradual rather than flash release of encapsulated drug represents a desired nanocarrier property[16]. We demonstrate that rapid drug release in combination with the enhanced accumulation of nanocarriers in the vessels near the tumour improves the drug therapeutic efficiency alongside with reduction of side effects. We believe that the demonstrated concept will change the existing approaches of anti-cancer nanomedicine design.

## Results
### Theoretical modelling of FlaRE drug delivery
To demonstrate that rapid drug release from nanocarriers to the tumour-surrounding vasculature (FlaRE drug delivery) is the most optimal for the cargo delivery to the tumour parenchyma, we developed a theoretical model illustrated in Fig. 1b and detailed in Supplementary Note 1. The FlaRE drug delivery mode is compared with slow drug release from nanocarriers, and free drug injection. We assume that a fraction of intravenously administered drug nanocarriers accumulates inside the capillaries of the target organ (e.g., lungs) and starts releasing low-molecular-weight cargo.

The released drug partly outflows from the target organ to the body compartments until the concentration equilibrium is established.

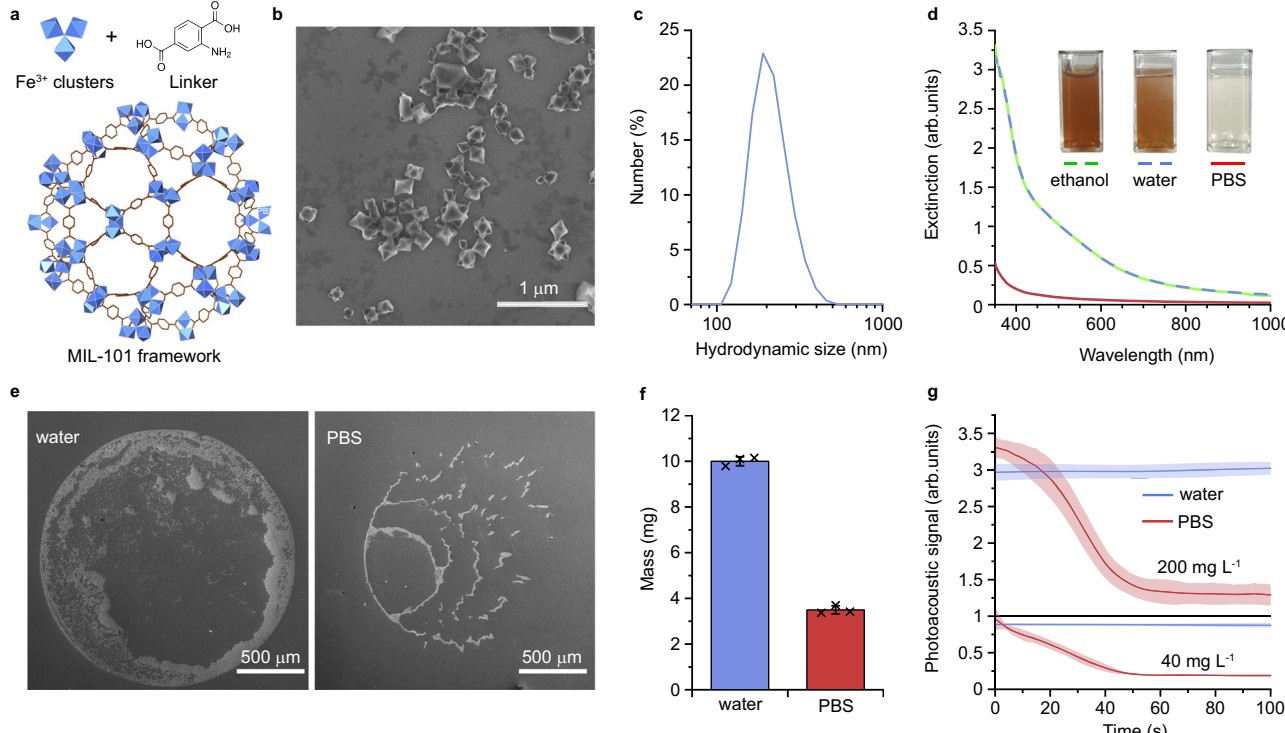

**Fig. 2 | Characterisation of MIL-101 nanoparticles and their degradation in PBS.**
**a** Schematic representation of the pore structure of MIL-101 nanoparticles (NPs).
**b** Scanning electron microscopy (SEM) micrograph of MIL-101 NPs, $n = 1$ sample.
Scale bar = 1 μm. **c** Hydrodynamic size distribution of MIL-101 NPs in water,
$n = 1$ sample, 3 repetitions. **d** Extinction spectra of MIL-101 NPs 24-h post incubation
in ethanol (green), water (blue), and PBS buffer (red). Inset—images of cuvettes with
NP solutions in dispersants, $n = 1$ measurement. **e** SEM images of two droplets of

MIL-101 NPs of the same volume and concentration 24-h post incubation in water
(left) and PBS (right), $n = 1$ experiment. Scale bars = 500 μm. **f** Mass analysis of MIL-
101 NP samples before (blue) and after degradation (red), $n = 3$ samples.
**g** Photoacoustic analysis of MIL-101 NP degradation kinetics in water (blue) or PBS
(red) at concentrations 200 mg L$^{-1}$ (top) and 40 mg L$^{-1}$ (bottom), $n = 3$ samples. In
**f**, **g** data are presented as mean values ± SD.

It crosses the capillary walls of the target organ, diffuses in the interstitium, and enters cancer cells. The process of the cargo transport and the rate of its cellular uptake is mediated by the drug binding to proteins. The bound drug cannot cross the capillary walls and its diffusion in the interstitium is slowed down. The peak concentration of the drug in cancer cells is the main determinant of the therapeutic outcome[17], and represents the target value of this model.

The concentration kinetics of the released molecular cargo in the interstitium and the tumour cells in the target organ is modelled using a hybrid compartmental model[18] and Krogh cylinder model[19]. The binding to proteins and cell internalisation of the drug is modelled by the Michaelis-Menten kinetics[20]. From the instant of the drug release from the carrier, its concentration in the capillaries, interstitium, and cancer cells grows. When the release exhausts, the intracapillary concentration drops and the drug diffusion is reversed to drug effusion from the tissue to the capillaries with ensuing excretion from the organism.

The release kinetics was modelled with a Heaviside function. We modelled four cases of equally dosed drug release: two – intravenous administration of free drug with rapid (3 min) and slow (3 h) infusion; and two – drug delivery by nanocarriers with rapid (3 min) and slow (3 h) drug release durations. A set of partial differential equations were solved numerically using Flex PDE 7.0 software, with relevant parameters collected in Supplementary Table 1.

Figure 1c, d show the radial distribution of the concentration of albumin-bound drug and its intracellular concentration, respectively, computed at several time points, when the rapid encapsulated drug delivery mode is considered. Note the relatively rapid drug concentration kinetics in the interstitium in comparison with that in cells. This trend of the interstitial drug concentration kinetics remains

similar in three other drug delivery modes considered (slow encapsulated drug release, rapid and slow drug infusion), although the rapid encapsulated drug release mode exhibits the highest drug concentration values (Supplementary Figs. 1, 2).

Figure 1e shows the kinetics of the intracellular concentration of drug (doxorubicin) computed at a fixed radial distance of 50 μm from the capillary wall. The rapid encapsulated drug release mode appears superior over the other considered drug delivery modes (Fig. 1e). The potential therapeutical benefit of the rapid release is based on the notion that the cell mortality monotonically increases versus the peak concentration of the drug, rather than the area under the curve (AUC)[18].

Our model prediction is in line with the reported results on drug delivery via externally triggered thermosensitive liposomes, such as a demonstrated 17-fold increase of the drug concentration in the tumour tissue[21]. However, this reported method was severely limited by local application of external radiation, and was suitable for treatment only of surgically accessible tumours but not for systemic treatment, which is of prime importance for cancer therapy.

**Development of rapidly degradable metal-organic frameworks**
In this study, we demonstrate therapeutic potential of the FlaRE concept experimentally after systemic administration of nanoparticles. To this aim, we developed rapidly degradable MIL-101 (Fe) metal-organic frameworks (MIL-101 NPs) as a carrier of chemotherapeutic drugs. The MIL-101 NPs consist of terephthalic acid derivatives and iron-based secondary building units. This material is characterised by a vast Brunauer–Emmett–Teller (BET) surface area up to 4500 m² g$^{-1}$ in addition to the giant pore size of 29–34 Å (Fig. 2a)[22,23]. This has enabled uploading, delivery, and release of large amounts of functional

molecules, including therapeutic cargo azidothymidine triphosphate, doxorubicin and cidofovir[22].

We produced MIL-101 NPs via standard solvothermal synthesis using iron (III) chloride and 2-aminoterephtalic acid as the reaction substrates[23]. The reaction yield was 73% for the iron conversion. As-synthesised MIL-101 NPs were of typical octahedral shape (Fig. 2b) with the hydrodynamic size of $215 \pm 60$ nm (Fig. 2c). The loading efficiency was analysed for various fluorophores, drugs, and bioactive molecules, ranging from 36% to 82% w/w (see Supplementary Table 2). As produced MIL-101 NPs were stable in distilled water for at least 24 h when transferred from ethanol (Fig. 2d).

Transfer of MIL-101 NPs to PBS resulted in rapid degradation of particles confirmed by UV-Vis spectroscopy and observed visually as change of the solution colouration from light brown to clear (Fig. 2d). This degradation was confirmed by SEM images of two droplets with equal initial content of MIL-101 NPs (Fig. 2e). The amount of the sample diminished after 24-h incubation in PBS, as clearly observable when compared to the sample incubated in water. This result was corroborated by $2.9 \pm 0.2$ times decrease of the dry weight of MIL-101 NPs after the incubation in PBS (Fig. 2f). Note the metal-organic framework degradation was incomplete, pointing to structural transformation of MIL-101 NPs rather than dissolution as the underlying mechanism. It required further investigative steps to elucidate this mechanism for implementation of the proposed drug delivery mode.

The kinetics of the MIL-101 NPs transformation was investigated by photoacoustic measurement technique due to the material strong optical absorption[24]. Upon the excitation with a 532-nm nanosecond pulsed laser, they generated a photoacoustic signal. Read out of the photoacoustic peak amplitude reported the particle concentration (Supplementary Fig. 3).

While the photoacoustic signal of the MIL-101 NPs in water remained stable over time, it dropped dramatically in 1 min when NPs were transferred to PBS, followed up by stabilisation at a constant level (Fig. 2g). We inferred that the MIL-101 NP degradation occurred during the $44 \pm 7$ s and $39 \pm 6$ s in PBS for the NP concentrations of 200 mg L$^{-1}$ and 40 mg L$^{-1}$, respectively.

Laser irradiation during the photoacoustic measurement can thermally or mechanically damage the nanoparticles enhancing their degradation[25]. Moreover, particle sedimentation during the measurement may lead to the underestimation of their quantity in environment. To reduce influence of these factors on the measurement results, we expanded the laser beam over the width of an 1-cm cuvette, at the same time lowering laser energy and enhancing irradiation volume. Supplementary Fig. 4 shows that the optical properties and photoacoustic responses showed no changes after 15-min laser exposure for intact MIL-101 NPs in water and for degraded MIL-101 NPs in PBS. It supports our notion that sedimentation and photodegradation had minimal influence on the kinetic analysis during the measurement period.

### Phosphate binding triggers MIL-101 degradation

First, we observed that the MIL-101 NP size distribution in PBS was downshifted, with the cut-off hydrodynamic diameter <300 nm from the sample, while the size fraction of 60-100 nm was enhanced (Fig. 3a). The mean size reduced from $159 \pm 70$ nm to $126 \pm 50$ nm. Second, we carried out in-beam SE (secondary electrons) and in-beam BSE (back-scattered electrons) scanning electron microscopy observations (Fig. 3b). The SE and BSE electron microscopy modes emphasise particle morphology and atomical density, respectively. The BSE signal is proportional to the mean atomic number ($Z$), so that the greater the quantity of high-$Z$ elements in the particle composition, the brighter is the BSE signal. The aqueous solution of MIL-101 NPs exhibited $Z_{MIL} < 14$, hence the BSE contrast of MIL-101 NPs was negative on the background of the silicon wafer ($Z_{Si} = 14$). The BSE contrast of PBS-incubated residual crystals of the NPs was noticeably brighter

(Fig. 3b, bottom left image), indicating the enhanced fraction of atomic elements with $Z > 14$, for instance, iron ($Z = 26$), the core cluster element of the metal-organic framework. Third, the electrophoretic scattering showed the reverse of the MIL-101 NP $\zeta$-potential from ($+37 \pm 9$ mV) to ($-26 \pm 4$ mV), following their degradation in PBS (Fig. 3c). Fourth, we measured the release of MIL-101 framework components in PBS by inductively coupled plasma mass spectrometry (ICP-MS) analysis and found that <0.2 % of iron was released from the nanoparticles in both aqueous and PBS solutions (Fig. 3d). The detection of another key component of MIL-101 framework, 2-aminoterephtalic acid, was carried out by UV-Vis spectroscopy. It revealed a 20-fold increase in the fluorescence of the linker in the supernatant, when the NPs sample was transferred to PBS (Fig. 3e). This result indicated that the linker was released to the solution. Taken together with the BSE results, the MIL-101 NPs degradation in PBS was accompanied by the loss of terephthalic acid, which led to the increase of the average atomic number of the degraded sample residue.

Fourier transformed infrared (FTIR) spectroscopy of MIL-101 NPs also favoured the linker substitution as a main mechanism of the nanoparticle transformation. The dried-up nanoparticle sample had a distinctive pattern in 2000–500 cm$^{-1}$ fingerprint region (Supplementary Fig. 5). The symmetric and asymmetric stretching vibrations of O = C–O manifested through the peaks at 1383 cm$^{-1}$ and 1576 cm$^{-1}$, respectively. The strong 1226 cm$^{-1}$ band corresponded to C–N stretching in 2-aminoterephtalic acid. The peaks at 765 cm$^{-1}$ and 590 cm$^{-1}$ were attributed to out-of-plane bending vibration of C–H in the aromatic ring and stretching vibration of Fe–O at the iron centres of MIL-101 NPs, respectively. The absence of a free carboxyl group peak near 1700 cm$^{-1}$ indicated that the as-prepared metal-organic frameworks were completely deprived of unbound 2-aminoterephthalic acid. After MIL-101 NP incubation in PBS, 1383-cm$^{-1}$ and 1576-cm$^{-1}$ peaks almost disappeared, while 590-cm$^{-1}$ Fe–O peak remained unchanged. This data supported the notion that the organic linker, but not iron release, was the underlying mechanism of the observed structural transformation. Besides, after the degradation of the nanoparticles, a broad peak at 1015 cm$^{-1}$ emerged, which probably attributed to the formation of Fe–O–P stretching.

The MIL-101 NP degradation in PBS resulted in a dramatic decrease of the particle surface area. The $N_2$ adsorption/desorption isotherm of the as-synthesised sample at 77 K featured type I characteristic, which corresponded to microporous solid (Fig. 3f). The BET surface area of this sample was measured as 2448 m$^2$ g$^{-1}$, in agreement with previous studies[23,26]. After the transformation, the surface area reduced 6.1-fold because of the pore collapse. Besides, the degraded MIL-101 NPs exhibited mesoporous-like type IV characteristics of $N_2$ isotherm featuring a hysteresis due to the microcapillary gas condensation[27]. The pore volume distribution curves by their average size showed decrease of the pore volume after the MIL-101 NP degradation (Supplementary Fig. 6).

X-ray diffraction (XRD) spectra of the dried MIL-101 NPs incubated in water and PBS were acquired, with the results presented in Fig. 3g. The diffraction pattern of the aqueous sample matched that of MIL-101 crystal structure pure phase[28]. The spectrum of the sample incubated in PBS displayed a complete loss of the crystallinity and emergence of $2\theta$ degrees broad peak at 28°, identified as amorphous iron phosphate species[29]. Since the terephthalic acid release might diminish the buffer acidity decreasing propensity of the hydroxyl attack, we further studied the particle degradation in PBS buffer at pH 8.5. The XRD pattern of MIL-101 NPs after 1-h incubation in alkaline buffer featured two broad peaks at 35° and 62° $2\theta$ degrees (Supplementary Fig. 7). These peaks corresponded to (110) and (115) planes of 2-line ferrihydrite[30], a low-crystalline iron oxyhydroxide species. This observation led us to speculate that the downstream degradation pathway of MIL-101 NPs through the full hydrolysis yielded ferrihydrite – an endogenous

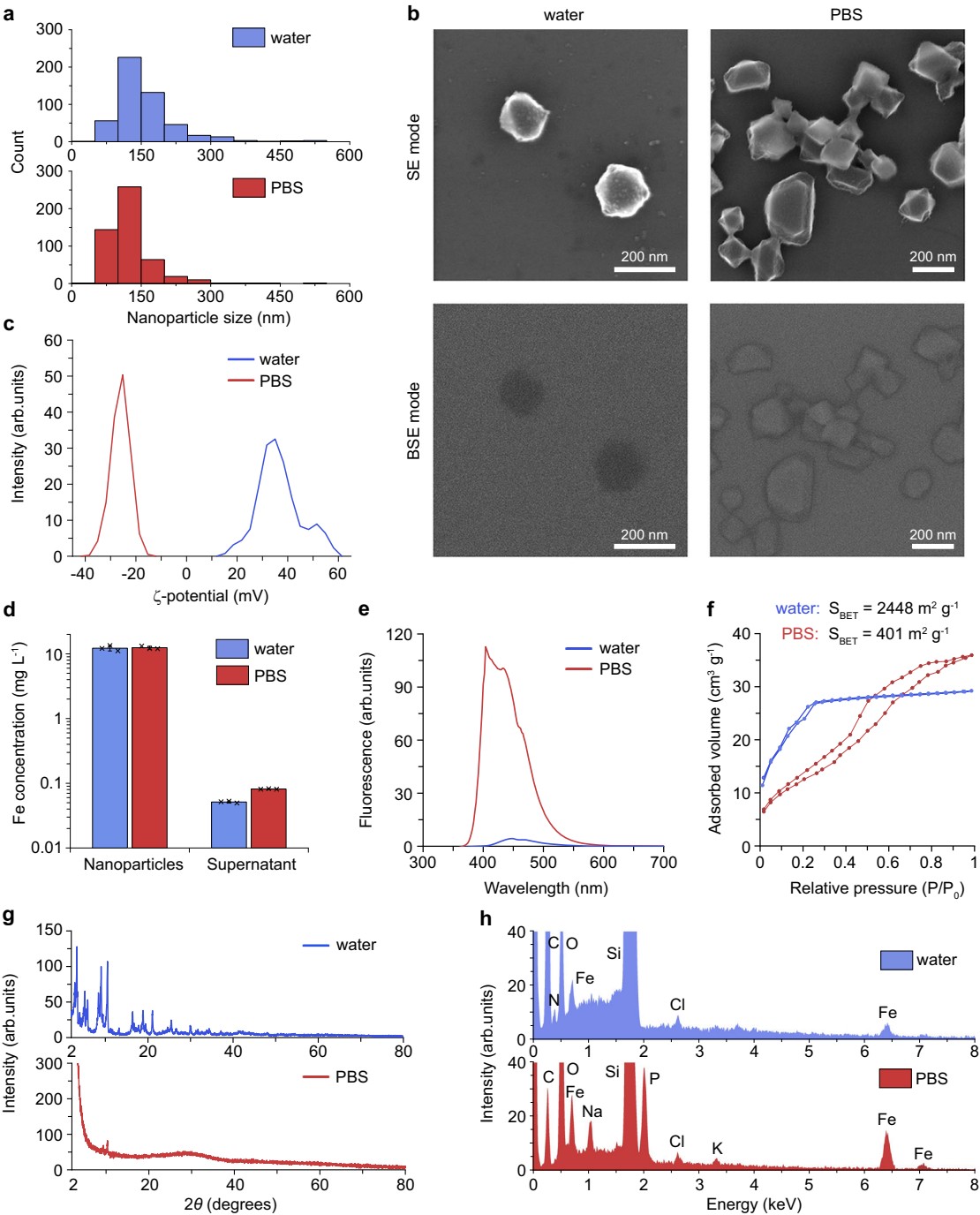

**Fig. 3 | MIL-101 nanoparticle degradation mechanism.** Results of MIL-101 nano-particle (NP) degradation analysis after 24-h incubation in PBS and water at the concentration 200 mg L⁻¹ are marked with red and blue, respectively (columns and lines). **a** Histograms of the particle size distribution based on scanning electron microscopy analysis ($n = 1$ experiment, 500 particles were analysed for each histogram). **b** Electron micrographs obtained using in-beam secondary electron (SE) mode (top row) and in-beam back-scattered electron (BSE) mode (bottom row) of MIL-101 NPs incubated in water (left) or PBS (right), $n = 1$ experiment. Scale bars = 200 nm. **c** The $\zeta$-potential distributions of nanoparticles, $n = 1$ experiment, 3 repetitions. **d** Iron concentration in the dry and supernatant fractions from the MIL-101 NPs incubated in water or PBS, $n = 3$ samples. Data are presented as mean values ± SD. **e** Fluorescence spectra of 2-aminoterephtalic acid released from MIL-101 NPs into the supernatant. Excitation−315/10 nm. $n = 1$ measurement. **f** Brunauer−Emmett−Teller N₂ adsorption/desorption isotherms of MIL-101 NPs. $n = 1$ measurement. **g** X-ray diffraction patterns of MIL-101 NPs. $n = 1$ measurement. **h** Energy-dispersive spectroscopy patterns of MIL-101 NPs. $n = 1$ measurement.

material, which was known to be deposited in the living organisms as the core of native ferritin[31].

The elemental composition of the MIL-101 particles incubated in water and PBS was analysed by energy dispersive spectroscopy (EDS). In comparison to the MIL-101 NP aqueous sample, the EDS spectrum of the MIL-101 NPs in PBS displayed a decrease of the carbon and nitrogen elemental peaks and an increase of the iron and oxygen peaks

amplitudes (Fig. 3h, Supplementary Table 3). It suggested the degradation of the metal-organic framework structure. The appearance of phosphorus, sodium, and potassium elements abundant in PBS, was explained by capturing $PO_4^{3-}$ ions by MIL-101 NPs and further association of monovalent metals either with phosphates or terephthalates. The supplementary study of the MIL-101 NP behaviour demonstrated that the increase of the phosphate concentration in the

buffer accelerated the particle degradation (Supplementary Fig. 8). This fact indirectly confirmed the key contribution of the phosphates to the particle degradation. The EDS data agreed with the previous reports, which indicated the possibility of rapid and selective binding of phosphates via the electrostatic attraction and linker exchange in Fe- and Al-based MIL-101 frameworks[32]. The same mechanism was proposed for the disassembly of other metal-organic frameworks, including MIL-100 (Fe) and ZIF-8 (Zn), although the kinetics of their degradation was significantly slower[33,34].

Based on the obtained results, the following mechanism of the MIL-101 (Fe) framework transformation in PBS can be proposed. Phosphate and hydroxide ions from PBS attack the iron clusters in MIL-101 NPs, effectively displacing terephthalic acid, which is released into the buffer while the iron core remains. It leads to the collapse of the pores, decreasing the volume and mass of the particles, and forming amorphous iron phosphate species. Further hydrolysis of iron phosphates leads to the emergence of a biocompatible low-crystalline form of ferrihydrite.

### MIL-101 NPs release molecular cargo prior to cellular uptake

To investigate the feasibility of MIL-101 NPs as a drug delivery vehicle we evaluated their cellular uptake and therapeutic efficiency in vitro. The observed rapid and irreversible collapse of the pores would result in a rapid release of therapeutic cargo without reabsorption. It was anticipated that serum proteins in culture medium and, more generally, plasma proteins in blood would slow down the MIL-101 NP degradation. This suggestion was evaluated by means of photoacoustic technique using $50\,g\,L^{-1}$ of albumin solution in PBS, which served as a model of the protein-rich environment. The time of complete MIL-101 NP degradation increased from $44 \pm 7\,s$ to $530 \pm 40\,s$ compared to the unsupplemented PBS (Fig. 4a). The observed deceleration of the MIL-101 NP degradation may be explained by the formation of a protein corona, which protected the particle core and prevented the organic linker substitution from the metal-organic frameworks.

Doxorubicin and rhodamine 123 were employed as a therapeutic drug and a fluorescent dye, respectively, for evaluation of the loading and release properties of the MIL-101 NPs. The weight per weight (w/w) loading efficacy reached $36.2 \pm 1.4\%$ for doxorubicin and $42.3 \pm 2.7\%$ for rhodamine 123. The kinetics of cargo release was equivalent to the kinetics of the particle degradation, with 90% of the cargo unloaded within the first 8 min and the remaining 10% – within 30 min measured from the incubation time point. The cumulative release reached 72.1% and 54.7% for doxorubicin and rhodamine 123, respectively.

Such flash release was expected to render the cytotoxic properties of the doxorubicin-loaded MIL-101 NPs equal to that of equivalent dose of free doxorubicin in vitro. We hypothesised the cells would uptake released doxorubicin from the incubation media rather than from the endocytosed particles. To test this hypothesis, we performed MTT cytotoxicity assay in NCI-H1299 human non-small cell lung carcinoma and B16-F1 murine melanoma cell lines. The choice of NCI-H1299 cells was justified by their universal use for in vitro studies of lung cancer in humans[35]. The B16-F1 cells have been successfully applied for establishing a murine model of extremely aggressive lung metastases[36].

In NCI-H1299 cells, the unloaded MIL-101 NPs had the $IC_{50}$ value as high as $65.3 \pm 6.2\,\mu g\,mL^{-1}$, which indicated the low toxicity of the nanomaterial. At the same time, the doxorubicin-loaded MIL-101 NPs had $IC_{50}$ value of $3.1 \pm 0.6\,\mu g\,mL^{-1}$ (or $930 \pm 180\,ng$ of doxorubicin), which was statistically equivalent to that of free doxorubicin with $IC_{50}$ value of $1.26 \pm 0.3\,\mu g\,mL^{-1}$ (Fig. 4c).

These observations were confirmed by the results of cytotoxicity assay in B16-F1 cells. The viability was >70% even in MIL-101 NP concentration as high as $125\,mg\,L^{-1}$. The $IC_{50}$ value for doxorubicin-loaded MIL-101 NPs was $1.02 \pm 0.28\,\mu g\,mL^{-1}$, which was equivalent to that of

free doxorubicin with $IC_{50}$ value of $418 \pm 106\,ng\,mL^{-1}$ (Supplementary Fig. 9). The difference in relative particle toxicity for the cell lines may be attributed to variations in their metabolism, repertoire of cell surface receptors, and subsequent changes in cell-nanoparticle interactions.

To analyse whether MIL-101 NPs were internalised only after they had been degraded, we carried out study using transmission electron microscopy (TEM). Their uptake by NCI-H1299 cells was observed 3 h and 24 h post incubation with the NPs found in early and late endosomes. At the same time, no NPs were found 15 min post incubation in the endosomes, as this incubation period was apparently insufficient for the particles to be internalised (Fig. 4d). To rule out that their sparse adherence to cell membrane was not an artefact caused by thorough washing of the specimen during sample preparation, we performed Perls staining of the iron-containing MIL-101 NPs. Few particles were observable on the cell surface 15 min post incubation followed by a single washing. This supported our hypothesis that cells took up doxorubicin from media, following the degradation of the doxorubicin-loaded MIL-101 NPs rather than after their endocytosis (Fig. 4e).

### MIL-101 NPs deliver drug deep in the lung tissue

Next, pharmacokinetics and toxicity of the MIL-101 NPs were evaluated in vivo. The biodistribution of the nanoparticles was analysed in female BALB/c mice by ICP-MS and histological analysis, following their intravascular administration via the tail vein dosed $25\,mg\,kg^{-1}$. The Perls staining of tissue slices showed the presence of the iron-containing particles in lungs and spleen as soon as 5 min after injection. These particles predominantly remained in the pulmonary capillaries 3 h post injection. Considering much faster drug release kinetics, we conjectured that most of the drug payload had been released while the MIL-101 NPs were immobilised inside the microcapillaries. Then we observed a gradual decrease in the MIL-101 NP quantity in the lungs with almost complete clearance in 2 weeks. At the same time, the iron content in the liver and spleen elevated during day 1 but was cleared in 2 weeks. (Supplementary Fig. 10). Generally, we did not observe significant iron accumulation in the heart and kidneys (Supplementary Fig. 10).

Quantitative ICP-MS analysis of iron concentration in the tissues confirmed rapid and effective pulmonary delivery of the MIL-101 NPs. Their accumulation in the lungs led to a 5-fold increase in the iron concentration in the tissue 15 min post injection, followed up by a gradual decrease over 3 weeks. We assume, that degraded MIL-101 NPs were cleared from the lungs and the residual iron content redistributed between the liver and spleen, two major organs of the mononuclear phagocyte system (Fig. 5b). Interestingly, the iron-content remained elevated in the lungs, heart, and kidneys even 3 weeks post injection according to ICP-MS analysis, although no traces of iron were visualised in the tissue slices after Day 14. This phenomenon may be explained by redistribution of the excess of the biogenic iron between the tissues and its storage as a part of ferritin complexes. In line with our observations, similar metabolic pathways of iron species have been reported for many other types of intravenously injected magnetic NPs in vivo[37,38]. The MIL-101 NPs were detected in the pulmonary capillaries 3 h post injection, indicating that discrete particles could be internalised and retained in endothelial cells for a long period. Note, the TEM analysis of the interaction between cells and the nanoparticles spoke in favour of this possibility (Fig. 4d).

We investigated the MIL-101 NP toxicity via histopathological analysis of the lungs, liver, spleen, kidneys, and heart. Generally, no dystrophic changes were found in these tissues. Only a slight increase in granular dystrophy of hepatocytes, in inflammatory infiltration of the stroma, and in the average number of Kupffer cells was observed in the liver on Day 3. However, these abnormal changes normalised by

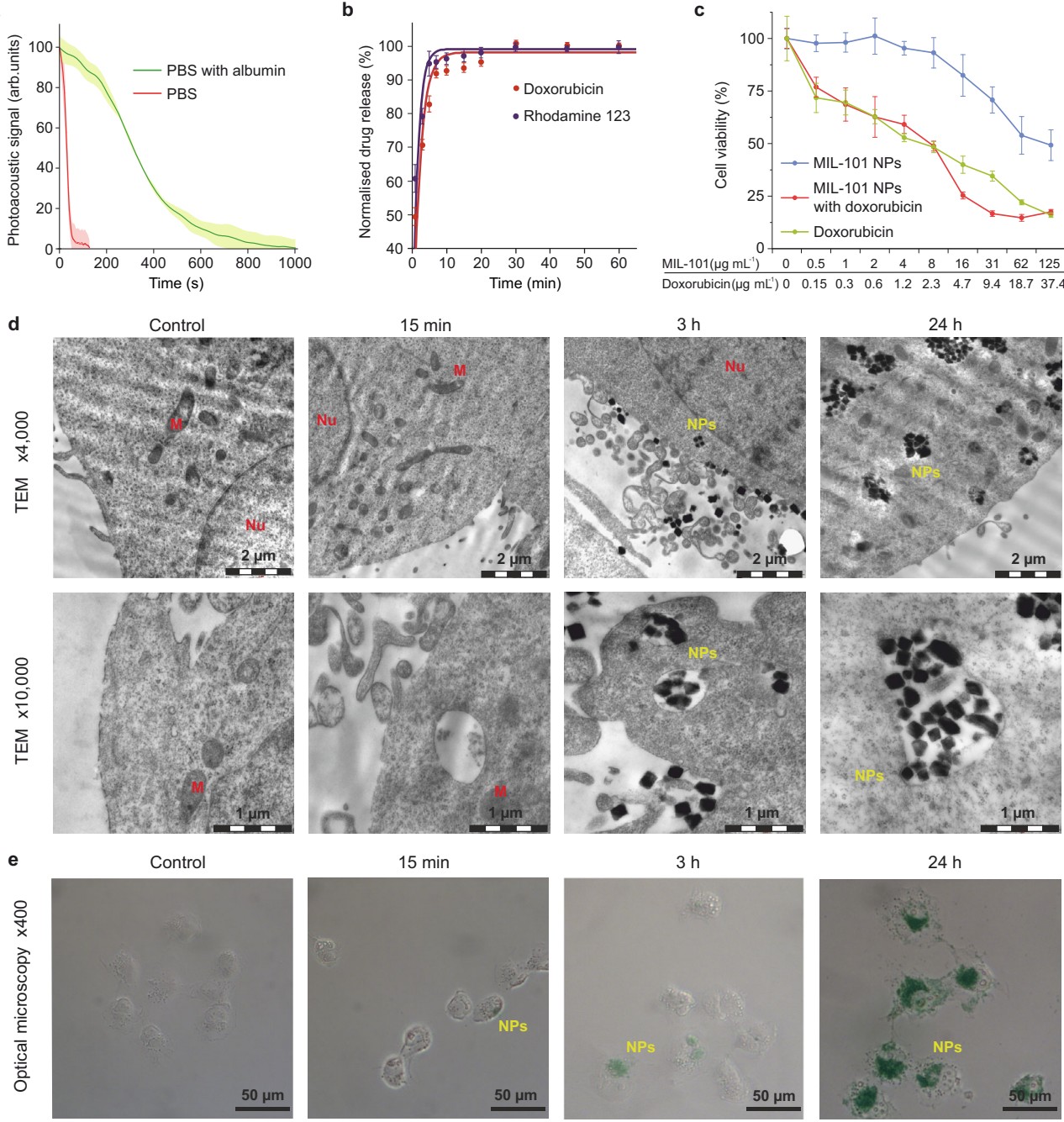

**Fig. 4 | The MIL-101 nanoparticles interaction with cells in vitro. a** The MIL-101 nanoparticle (NP) degradation kinetics in PBS (red) and PBS supplemented with $50\,g\,L^{-1}$ of albumin (green), $n = 3$ samples. **b** The normalised release kinetics curves for rhodamine 123 (purple) and doxorubicin (red) from the MIL-101 NPs in serum. $n = 3$ samples. **c** MTT cytotoxicity assay of MIL-101 NPs (blue), doxorubicin-loaded MIL-101 NPs (red) and free doxorubicin (green) with NCI-H1299 cells. The cell viability is presented as % ratio normalised to the non-treated control cells, $n = 5$ samples analysed per data point. **d** Representative images of the MIL-101 NP uptake by NCI-H1299 cells 15 min, 3 h and 24 h post incubation. Top−transmission electron microscopy (TEM) images showing the MIL-101 NP uptake and all stages of endocytosis 3 h and 24 h post incubation, and no uptake 15 min post incubation. Scale bars = $2\,\mu m$. Bottom−enlarged TEM images of the MIL-101 NP uptake indicating the absence of particle internalisation in 15 min. $n = 3$ samples. Scale bars = $1\,\mu m$. Nu and M denote nucleus and mitochondrion, respectively. **e** Optical microscopy images of the MIL-101 NPs stained with Perls Prussian Blue in NCI-H1299 cells. The MIL-101 NP binding was clearly observable with blue colour. $n = 3$ samples. Scale bars = $50\,\mu m$. In **a**−**c** data are presented as mean values ± SD.

Day 14. The structure of other organs was normal, without any pathological changes. Therefore, we conjectured that the intravenously administered MIL-101 NPs were safe (Supplementary Fig. 11, Supplementary Table 4).

Once the predominant accumulation of the MIL-101 NPs in the lung capillaries was established, we turned to investigation of the drug transport to surrounding tissues to test the theoretical modelling

predictions. MIL-101 NPs were loaded with rhodamine 123, fluorescent dye which release profile was similar to that of doxorubicin, while the contrast on autofluorescence background of the tissue was enhanced. Data fitting with the Weibull model of the cumulative release, $C = a(1 - e^{-kt})$ showed that the time constants $(k)$ were $0.48\,min^{-1}$ and $0.69\,min^{-1}$ for doxorubicin and rhodamine 123, respectively (Fig. 4b).

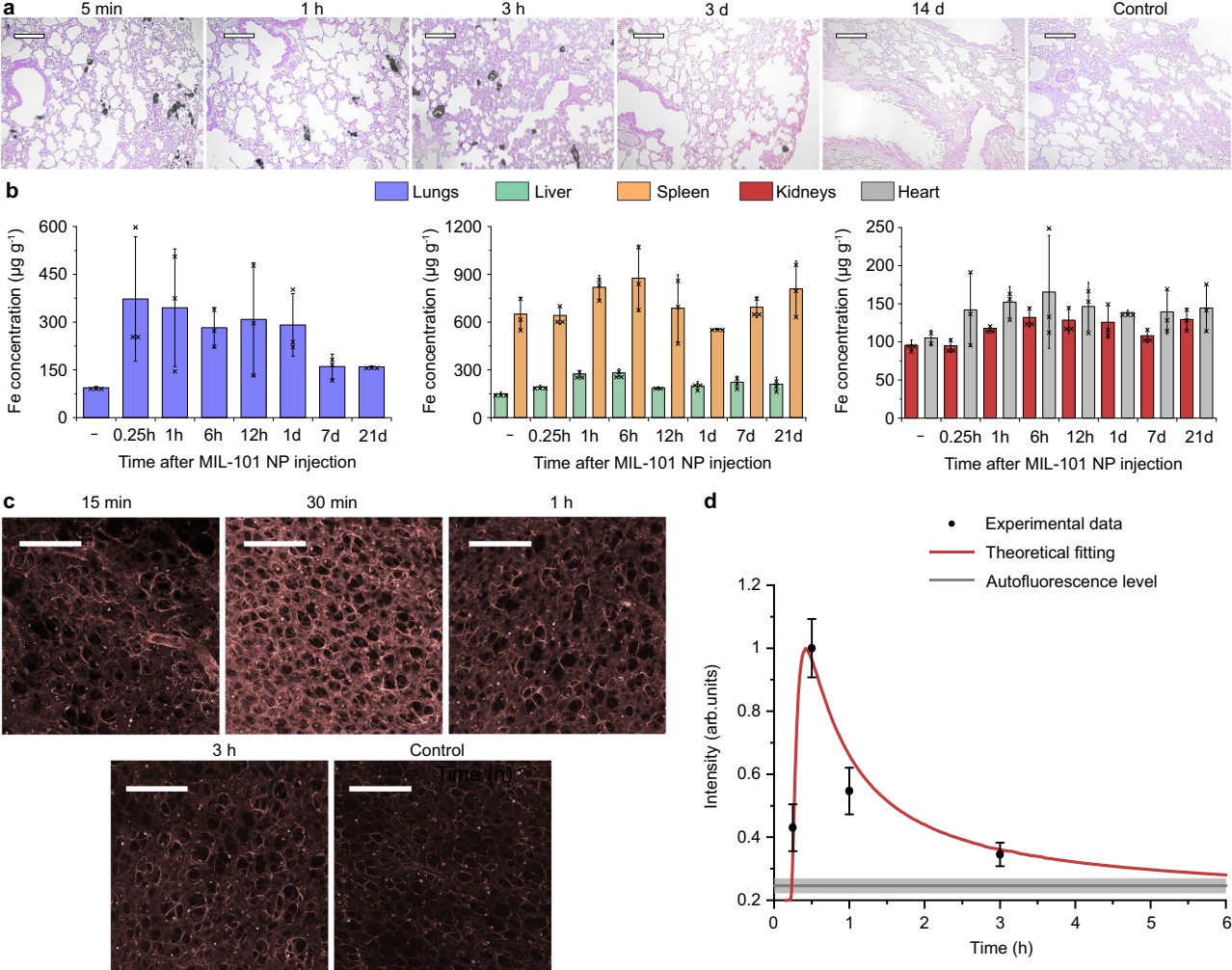

**Fig. 5 | Investigation of the drug delivery to the lungs using MIL-101 nano-particles. a** Histological evaluation of the lung tissue at several time points, following the MIL-101 NP injection. The tissue slices were stained with eosin and Perls Prussian blue. $n = 3$ mice. Scale bars = 50 μm. **b** Evaluation of the iron concentration in the main organs after the MIL-101 NP injection: lungs (blue), liver (green), spleen (orange), kidneys (red), heart (grey). $n = 3$ mice. **c** Representative confocal images of rhodamine 123 distribution in the lung tissue at several time points following the MIL-101 NP injection. $n = 3$ mice. Fluorescence excitation/emission, 488/515-575 nm. Scale bars = 250 μm. **d** Kinetics of the mean fluorescence intensity of the lung tissue. Data were fitted using the developed theoretical model (red line). Grey line marks the mean ± SD tissue autofluorescence level. $n = 3$ mice. In **b**, **d** data are presented as mean values ± SD.

The formulated dye-loaded MIL-101 NPs were intravenously administered at a dose of 25 mg kg⁻¹. The mice were sacrificed at time points ranging from 15 min to 3 h post injection and the lungs were collected for investigation. The dye distribution in unprocessed interstitial tissue was immediately imaged by laser-scanning fluorescent confocal microscopy. We observed that the fluorescence signal of the tissue gradually increased for the first 30 min after the injection, followed by an exponential decrease over a 3-h period. The acquired data were compared with our theoretical model, using an algorithm described in Supplementary Note 2. In brief, we fitted the kinetics of the bound drug concentration, obtained in our theoretical model (Fig. 1c) with a Mathematica 13.1 software, to an analytical function, where the distance from a capillary was used as a fitting parameter. This function well fitted the experimental data of the dye distribution in the lungs (Fig. 5d), providing further validation of the FlaRE drug delivery model.

### Doxorubicin-loaded MIL-101 NPs effectively cure lung metastases

To evaluate whether the enhanced permeability of the tissue was therapeutically beneficial, we established an experimental B16-F1 melanoma lung metastasis model, which was extremely aggressive and associated with a high mortality rate[36]. The evaluation has been carried out in two murine models of melanoma. The first one was early-stage metastases with melanoma cell clusters treated on Day 1 after the cell administration. The second model was late-stage metastases with observed cancer cell extravasation from the capillaries to the surrounding tissue and the treatment started on Day 7 after the cell administration.

For the early-stage metastasis model[39], $1 \times 10^5$ B16-F1 cells were administrated in female C57Bl/6 mice via the tail vein. Next, the tumour bearing mice were administered with doxorubicin-loaded MIL-101 NPs dosed 10 mg kg⁻¹ on Days 1, 3, and 5. The control groups received free doxorubicin in the dose equivalent to the one loaded to the nanoparticles or PBS in the same regimen. The mice were sacrificed on Days 10 and 15 post induction of the lung metastases. The lungs were harvested, metastases were counted and sized under a dissecting microscope (Fig. 6a).

Kaplan-Meier survival analysis showed that the treatment with doxorubicin-loaded MIL-101 NPs was significantly more beneficial compared to the other groups ($P = 0.004$). The median survival time was 27 and 28 days for doxorubicin and PBS treated groups,

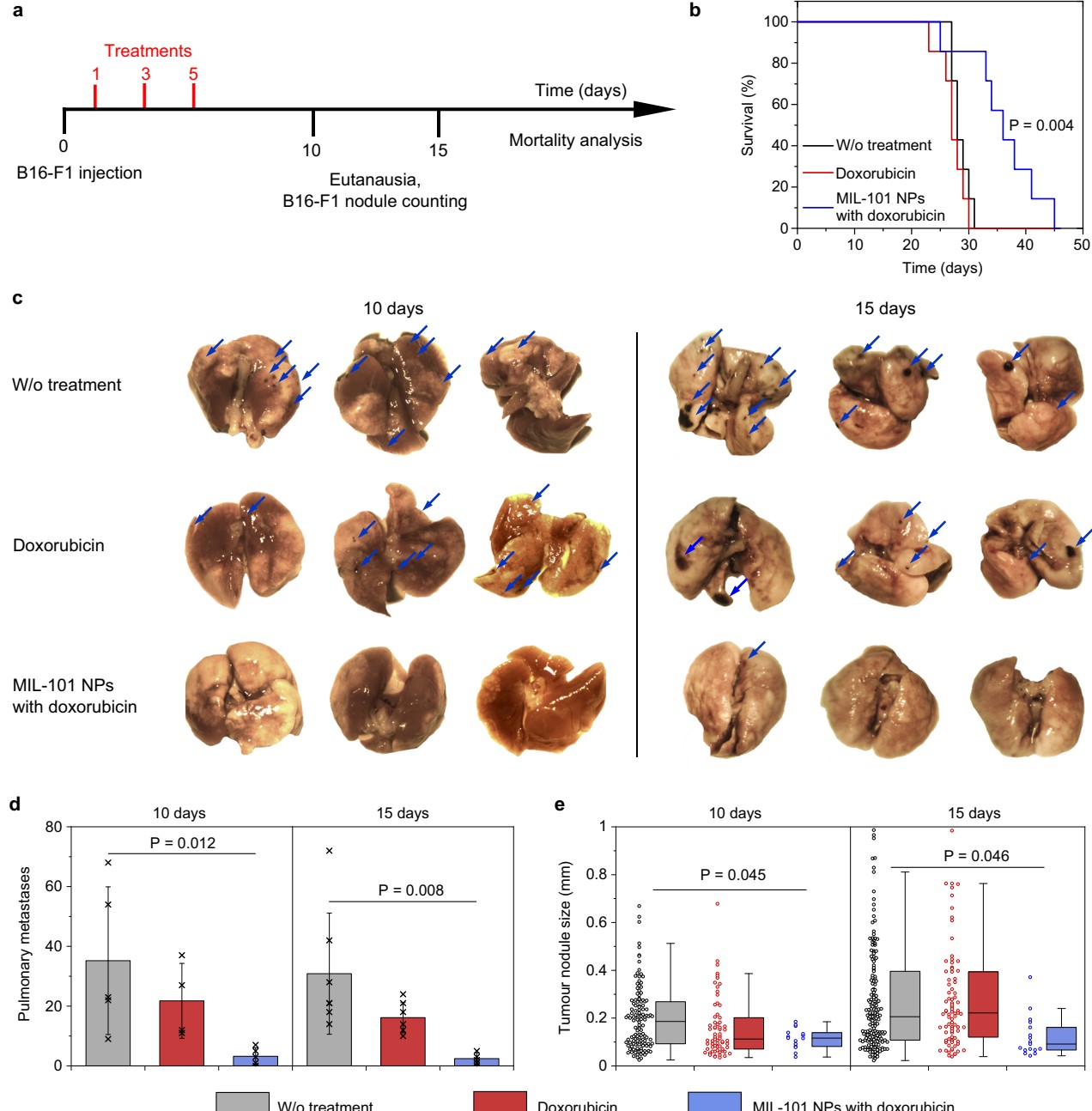

**Fig. 6 | Doxorubicin-loaded MIL-101 NPs effectively treat melanoma in early-stage pulmonary metastasis model. a** Treatment schedule. **b** Kaplan-Meier survival curves of mice after the treatment with PBS (black), doxorubicin (red) or doxorubicin-loaded MIL-101 NPs (blue). $n = 7$ mice per group. Two-sided log-rank test was used for statistical analysis. **c** Representative images of lungs, collected on Day 10 (left) or 15 (right) post cancer cell administration. Mice were treated with PBS (top line), doxorubicin (middle line) or doxorubicin-loaded MIL-101 NPs (bottom line). Blue arrows point to the melanoma metastases. $n = 3$ mice per group. **d** Quantification of the surface metastases in the harvested lungs on Day 10 and Day 15 after the treatment with PBS (grey), doxorubicin (red) or doxorubicin-loaded MIL-101 NPs (blue). Data are presented as mean values ± SD. **e** Analysis of the nodule size in the harvested lungs on Day 10 and Day 15 after the treatment with PBS (grey), doxorubicin (red) or doxorubicin-loaded MIL-101 NPs (blue). The boxplots represent median values, interquartile ranges and Tukey whiskers with individual data points superimposed. In **d, e** $n = 4$ mice for doxorubicin group on Day 10, $n = 5$ mice for the other groups on Day 10; $n = 6$ mice for all groups on Day 15. In **e** following metastases quantity was analysed in the mice lungs: 126, 66 and 15 for PBS, doxorubicin and doxorubicin-loaded MIL-101 NPs treated groups, respectively, on Day 10; 199, 85 and 17 for PBS, doxorubicin and doxorubicin-loaded MIL-101 NPs treated groups, respectively, on Day 15. One-way ANOVA with Tukey post-hoc test was used for statistical analysis.

respectively. On the other side, treatment with doxorubicin-loaded MIL-101 NPs increased median survival to 36 days (Fig. 6b). The number of surface metastasis nodules in the MIL-101 NP treated group decreased dramatically, as shown in representative lung images (Fig. 6c). Treatment with doxorubicin caused 1.9-fold and 1.6-fold decrease of the metastasis quantity on Days 10 and 15, respectively. On the other hand, after the treatment with doxorubicin-loaded MIL-101 NPs, the number of melanoma nodules decreased by 12.7-fold and 11-fold on Days 10 and 15 (Fig. 6d). We also observed decrease in melanoma nodule sized after the treatment with MIL-101 NPs. On Day 10, the median size of the metastases was 113 μm, 117 μm and 186 μm for doxorubicin, doxorubicin-loaded MIL-101 NPs and PBS-treated groups, respectively. However, on Day 15 the median metastasis size in the doxorubicin and PBS treated groups were comparable – 222 μm and

206 μm, while it decreased to 92 μm in the group treated with doxorubicin-loaded MIL-101 NPs. Histopathological analysis of the lungs also revealed a significant reduction of the size of the pulmonary metastases in the group treated with the doxorubicin-loaded MIL-101 NPs (Supplementary Fig. 12).

Interestingly, while free doxorubicin in 3.6 mg kg⁻¹ concentration reduced the quantity of metastases (Fig. 6d), it did not significantly affect the mice survival time (Fig. 6b). Our observation correlates with the previous studies which demonstrated no effect of high doses of doxorubicin on prolongation of life duration of mice[39,40]. For example, administration of free doxorubicin in 5.4 mg kg⁻¹ dose, which is 1.5 times higher than in present study, increased median survival time in murine melanoma model from 29 to 32 days only[39]. We assume that this effect may be explained by the low influence of free doxorubicin on proliferation rate of melanoma cells in vivo. On Day 15 after cell administration, we have not observed any decrease in the mean size of metastases in the free doxorubicin-treated group (Fig. 6e). I.J. Fidler have shown that only a small number of initial micrometastases evolve to a life-threatening macrometastases stage[41]. In case of present study this notion may be interpreted as follows. The administration of free doxorubicin may reduce the number of smaller metastases on their initial development stage; however, the rapid growth of the remaining ones would eventually lead to a death outcome in a short time. Encapsulation of the drug into MIL-101 NPs increases doxorubicin delivery efficacy, which manifests both in the decrease of metastasis size on Day 15 and in the prolongation of mice survival. It validates the proposed FlaRE delivery concept for treatment of disseminating melanoma at the early stages of metastases development.

For the late-stage metastasis model[39], $1 \times 10^5$ of B16-F1 cells were administered in female C57Bl/6 mice via the tail vein (Fig. 7a). Histological evaluation of the lungs harvested from 3 random mice on Day 7 demonstrated presence of numerous micro-metastases of 95 ± 40 μm size. The tumour cells started extravasation from the capillaries to the surrounding lung tissue (see Fig. 7b and Supplementary Fig. 13). Doxorubicin-loaded MIL-101 NPs were systemically administered on Days 7, 9, and 11 via the tail vein in a dosage of 10 mg kg⁻¹. Control groups were treated with free doxorubicin in the equivalent loaded dose or PBS in the same regimen, respectively. The animals were sacrificed on Day 18, their lungs were collected, metastases were counted and sized under a dissecting microscope.

In general, doxorubicin-loaded MIL-101 NPs were less effective for a late-stage metastases treatment than for early-stage one. Nevertheless, the MIL-101 NPs still showed better therapeutic efficacy than free doxorubicin. The number of melanoma nodules in the groups treated with doxorubicin-loaded MIL-101 NPs and free doxorubicin decreased 4.3-fold and 1.5-fold, respectively, compared to the PBS-treated negative control. The median size of the nodules decreased from 482 μm in the PBS-treated group to 317 μm for the doxorubicin-loaded MIL-101 NP treated group and to 316 μm for the free doxorubicin treated group. We suggest that for the late-stage metastasis model, MIL-101 NPs retain the better treating efficiency compared to the free drug, which supports the FlaRE drug delivery model.

Histological study showed that the systemic treatment of metastatic tumours with the doxorubicin-loaded MIL-101 NPs induced no pathological changes in the lungs and other major organs, except for an increase in the number of siderophages in the spleen (Supplementary Note 3). In addition, free doxorubicin and the drug loaded MIL-101 NPs induced reduction of the number of Kupffer cells in the liver (by 38% and 43%, respectively), decreasing them almost to the normal values of the healthy control animals (Supplementary Table 5). On the other hand, the administration of free doxorubicin showed slight toxicity. The emphysema, hyperaemia, haemorrhages, and bronchial spasm in the lungs were observed. The number of megakaryocytes in the spleen also increased, as compared to that of the control group (Supplementary Fig. 14, Supplementary Tables 6,7). The results of the histopathological analysis indicated that treatment with the drug-loaded MIL-101 NPs enhanced the therapeutic window of doxorubicin.

## Discussion

In this communication, we introduced a powerful FlaRE delivery strategy to cure metastatic tumours, which relied on the (i) rapid nanoparticle delivery to the vasculature of the injured organ and (ii) minutes-scale degradation of drug-loaded nanocarriers in the vessels with flash release of the drug. This condition was not obvious but essential to ensure efficient extravasation of small molecules across the capillary walls and ensuing deep permeation throughout the interstitium driven by the high gradient of the intracapillary drug concentration. The drug retention in the interstitium was facilitated by the binding of drugs to proteins and proved essential for the attainment of peak concentrations in the pharmacological targets. Our theoretical modelling predicted the optimal drug release as rapid as ranging from 10 min to 3 h. The efficiency of the systemically administered free drug was compromised by the inferior delivery to the target tissue when compared to the encapsulated drug, alongside with the adverse effect of the plasma protein binding that slowed down its permeation.

Our strategy expands the repertoire of potential drug carriers by redefining their advantageous properties to prioritise rapid drug release kinetics. Unmodified MIL-101 (Fe) frameworks represent an example of such unprecedented drug nanocarrier. Uncoated MIL-101 NPs exhibited the degradation time of 44 ± 7 s to 530 ± 40 s in PBS and protein-rich phosphate solution, respectively. The drug release rate of 0.48 min⁻¹ would rule out these nanoparticles as drug carriers in the existing paradigm of drug delivery but made these optimal for the demonstration of proposed FlaRE approach efficacy.

Suggested approach of FlaRE delivery differs from the EPR-based drug delivery strategies, which currently dominate nanomedicine. This approach is based on a rapid degradation of drug-loaded nanocarriers accumulated in the targeted vasculature and does not require extravasation of nanoparticles for effective drug delivery. In this study, the efficacy of FlaRE approach was experimentally demonstrated in treatment of metastatic tumours in lungs. MIL-101 NPs were preferentially trapped in the lung vasculature (~60% ID), likely due to their increase in size and to the change in their ζ-potential to the positive values after the phosphate binding in the bloodstream. Similar approach for targeting of an entire diseased organ has been demonstrated previously for selective organ targeting (SORT) nanoparticles and for RBC-hitchhiking technology[42,43]. Combination of rapidly degradable nanoparticles with specific targeting can further increase efficacy of FlaRE approach. The existing examples include active targeting of the inflamed vessel endothelium in lungs and brain due to NP anchoring to ICAM-1 and PECAM-1 receptors[44].

The proposed approach might be theoretically generalised for treatment of primary tumours, if the passive accumulation of particles in the tumour microcapillaries may be demonstrated. The reduced blood speed in the tumour vessels was one of the key mechanisms of the particle adhesion to the endothelium reported by Decuzzi, et al. (5% ID g⁻¹)[45] and Parakhonskiy, et al. (30% ID g⁻¹)[46]. Active endothelial targeting of the solid tumour vessels has been also reported[47]. The accelerated angiogenesis provides several other targeting options – anti-fibronectin EDB domain antibody fragment L19 as well as many RGD and NGR peptides binding integrins to the angiogenic endothelium may be employed[48].

Moreover, since the requirement of the extravasation (essential for EPR-based drug delivery strategies) is abrogated in favour of maximum delivery efficiency to the vasculature of the target tissue, the dimensional range of optimal drug carriers may be expanded towards sub-micron sizes. This provides a new view on nanomedicine design. For example, sub-micrometre discoidal particles have been reported

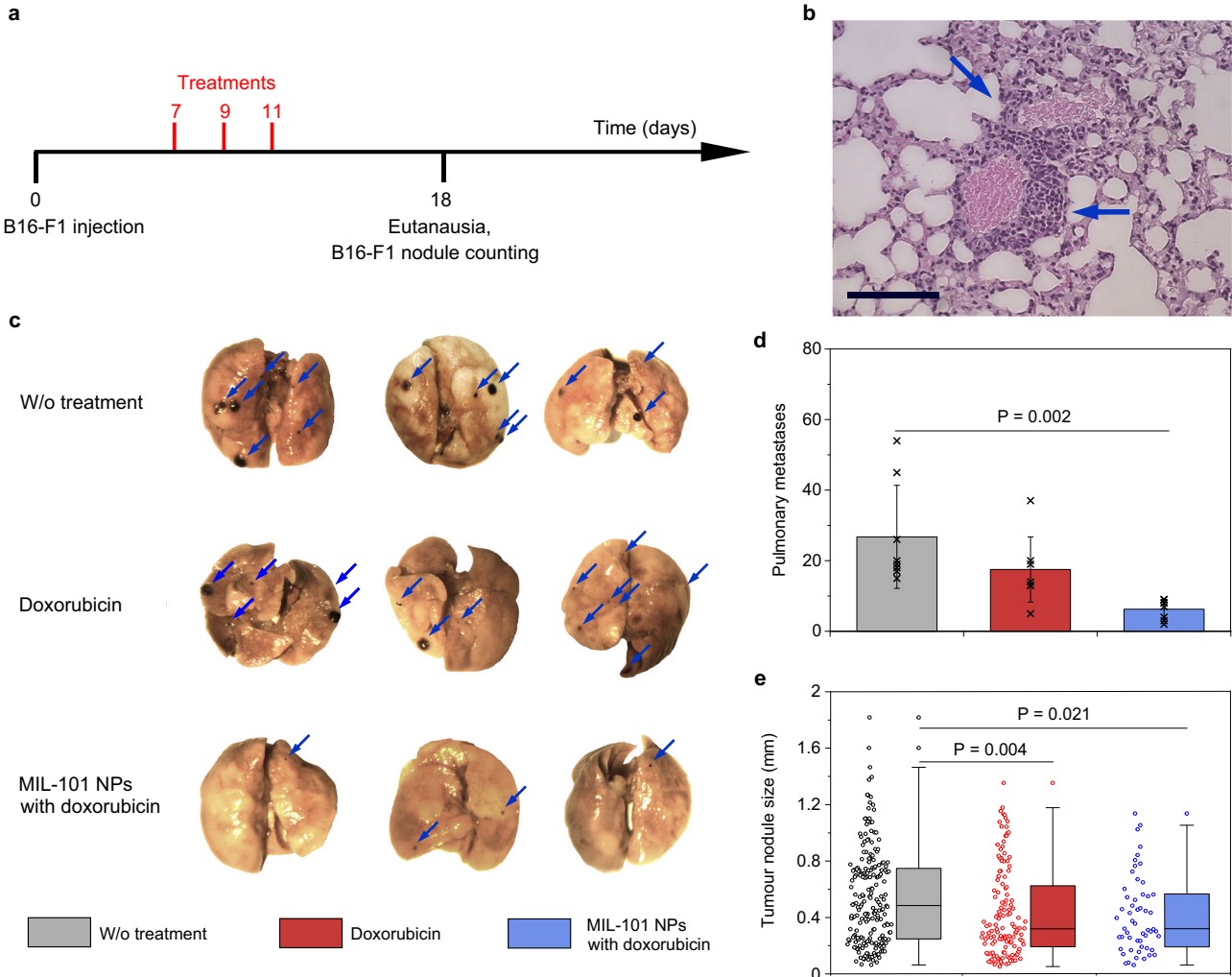

**Fig. 7 | Doxorubicin-loaded MIL-101 NPs effectively treat melanoma in late-stage pulmonary metastasis model. a** Treatment schedule. **b** Representative histological images, showing extravasation of B16-F1 melanoma cells from blood vessel to surrounding tissue. Scale bar = 100 μm. *n* = 3 mice. **c** Representative images of the lungs, collected on Day 18 post cancer cell administration. Mice were treated with PBS (top line), doxorubicin (middle line), or doxorubicin-loaded MIL-101 NPs (bottom line). Blue arrows point to melanoma metastases. *n* = 3 mice. **d** Quantification of the surface nodule metastases in the harvested lungs after the treatment with PBS (grey), doxorubicin (red) or doxorubicin-loaded MIL-101 NPs (blue). Data are presented as mean values ± SD. **e** Analysis of the nodule size in the harvested lungs after the treatment with PBS (grey), doxorubicin (red) or doxorubicin-loaded MIL-101 NPs (blue). The boxplots represent median values, interquartile ranges and Tukey whiskers with individual data points superimposed. In **d**, **e** *n* = 8 mice per group. In **e** following metastases quantity was analysed in mice lungs: 212, 136 and 55 for PBS, doxorubicin and doxorubicin-loaded MIL-101 NPs treated groups, respectively. One-way ANOVA with Tukey post-hoc test was used for statistical analysis.

to be effective as intravascular carriers to maximise accumulation in the target organ[45,47].

The introduced FlaRE delivery approach implemented with the MIL-101 NPs may be used for the treatment of lung fibrosis, infections, and other diseases, alongside with lung cancers, taking advantage of the enhanced permeation of drugs in the lung tissue. This can be worthwhile for post-Covid-19 treatment of fibrotic lesions in the lung tissue whose widespread occurrence has become evident during the pandemic.

## Methods
### Animals
Female BALB/c (10–14 weeks old) and C57BL/6 (8–12 weeks old) mice of 18–22 g weight were used in the experiments. Animals were obtained from Pushchino Animal Facility (Pushchino, Russia) and maintained in a vivarium at Shemyakin-Ovchinnikov Institute of Bioorganic Chemistry. All experimental procedures were approved by the Institutional Animal Care and Use Committee (protocol № 240) of

Shemyakin-Ovchinnikov Institute of Bioorganic Chemistry. The mice were housed under a 12-h light/dark cycle at room temperature (23–26 °C) and humidity (40–60 %) with ad libitum access to food and water. Animals were anaesthetised with an intraperitoneal injection of mixture of Zoletil and Rometar in a dose 40 mg kg⁻¹ and 1.6 mg kg⁻¹, respectively, prior to any drug injections. The maximal total volume for all tumours in mice, permitted by ethical protocol, was 2000 mm³. This value was not reached during the experiments.

### Materials
All chemicals were obtained from the following suppliers. Sigma-Aldrich (USA): 2-aminoterephtalic acid, iron (III) chloride hexahydrate, dimethylformamide (DMF), bovine serum albumin (BSA), dimethyl sulfoxide (DMSO), acetone, ethanol, potassium hexacyanoferrate (II) trihydrate, hydrochloric acid, nitric acid, doxorubicin hydrochloride, rhodamine 123, paraformaldehyde, sucrose, glutaric dialdehyde, sodium cacodylate trihydrate, methylthiazolyldiphenyl-tetrazolium bromide (MTT). Honeywell (USA): osmium tetraoxide, lead (II)

citrate tribasic trihydrate, uranyl acetate dihydrate. Fluka (Switzerland): araldite and EPON epoxy resins. Thermo Fisher Scientific (USA): Dulbecco's modified Eagle's medium (DMEM) and fetal bovine serum (FBS). PanEco (Russia): L-glutamine and gentamicin. Virbac (France): Zoletil. Bioveta (Czech Republic): Rometar.

NCI-H1299 human non-small cell lung carcinoma cell line from ATCC collection was generously gifted by Prof. Nikolai Barlev, Institute of Cytology of Russian Academy of Sciences. B16-F1 cell line from ATCC collection was generously gifted by Prof. R. I. Yakubovskaya, Moscow Hertsen Research Institute of Oncology, Russian Ministry of Health Care. Prior to the experimental use the cell lines were negatively tested for mycoplasma infection.

### Synthesis of the MIL-101 (Fe) metal-organic frameworks

82.6-mM 2-aminoterephthalic acid and 154-mM $FeCl_3 \cdot 6H_2O$ were dissolved in DMF. Then, the solution was heated at 120 °C for 1 h. The obtained nanoparticles were separated by centrifugation at 10 000 g for 15 min, washed 3 times with hot DMF, and 3 times with hot ethanol to remove unreacted chemicals from the pores.

### Nanoparticle characterisation

Hydrodynamic size (number distribution) and $\zeta$-potential of nanoparticles were measured using Malvern Zetasizer Nano ZS (Malvern Instruments, UK) with Zetasizer Software 7.11. All measurements were carried out in distilled water (for size analysis) or 10 mM NaCl (for $\zeta$-potential analysis). The degradation of the MIL-101 NPs was realised by incubation for 1 h at room temperature in PBS (200 mg $L^{-1}$), if not stated otherwise.

UV-Vis and fluorescent spectroscopy measurements were carried out using Infinite M1000 microplate reader (Tecan, Switzerland) operating with Magellan 7.2 software. To evaluate the release of 2-aminoterephtalic acid or preloaded small molecules after the MIL-101 NP degradation, the solution was centrifuged at 10,000 g, the supernatant pipetted out and analysed. The fluorescence spectrum of the 2-aminoterephtalic acid was measured in the spectral range of 350 – 650 nm upon the excitation at 315/10 nm. The absorbance or fluorescence of the other small molecules was measured at their characteristic wavelengths.

The iron release was quantified via inductively coupled plasma mass-spectrometry using NexION 2000 spectrometer (Perkin Elmer, USA) operating with Syngistix 2.4 software. Non-degraded and degraded particles were dissolved in the concentrated nitric acid. $^{57}Fe$ peak was used for the analysis.

Prior to the XRD, FTIR, and SEM characterisation, the MIL-101 NPs were washed thrice with hot ethanol and dried under vacuum. FTIR spectra were measured using FT-801 spectrometer with a diamond thermal cell of frustrated total internal reflection (Simex, Russia). Spectrometer was operated with ZaIR 3.5 software. The spectra were measured in 5700 – 470 $cm^{-1}$ range. XRD patterns were recorded using LabX XRD-6000 diffractometer (Shimadzu, Japan) with a CuKα radiation source ($\lambda = 1.54182$ Å) at the operating voltage of 40 kV (30 mA). The $2\theta$ range of 2–120 degrees was measured with a step size of 2 degrees $min^{-1}$.

The nitrogen adsorption and desorption isotherms were recorded at 77 K with Quantachrome NOVA 4200e analyser (Quantachrome Instruments, USA), operating with NovaWin 11.3.0.5 software. Nanoparticle powder was degassed under vacuum at 70 °C for 12 h. The Brunauer-Emmett-Teller (BET) method was used for the specific surface area quantification. The pore volume distribution of the MIL-101 NPs was analysed using Barrett, Joyner and Halenda (BJH) method.

The MIL-101 NP morphology, size, and composition were characterised by scanning electron microscopy (SEM) system MAIA3 (Tescan, Czech Republic) coupled with an energy dispersive spectrometry (EDS) detector (X-act, Oxford Instruments, High Wycombe, UK). Tescan MAIA3 Control Software 4.2.19.1 was used for microscope

operation, while Aztec One 3.5 software was used for collecting and analysing of EDS spectra. The MIL-101 NPs in water were dropped onto a silicon grid and dried up under the air. Electron micrographs were obtained using in-beam secondary electrons or in-beam back scattering electron detection modes and the accelerating voltage of 15 kV. The particle size distribution was analysed using ImageJ 1.8.0 software, with at least 500 particles measured. The Si peak was excluded from the EDS spectral analysis due to its presence in the grid.

### Photoacoustic measurements

The photoacoustic signal was acquired using our lab-built system[24]. The MIL-101 NP aqueous colloids were illuminated with a nanosecond pulsed Nd:YAG Quantas Q1B laser (Quantum Light Instruments, Lithuania). The 2nd harmonic at 532 nm was employed with pulse repetition rate of 20 Hz, pulse duration of 5 ns and energy per pulse of 5 mJ. The laser beam was expanded to fill an 1-cm cuvette width by a diverging lens, to minimise nanoparticle damage by the laser irradiation. A silicon DET36A/M 350–1100 nm photodetector (ThorLabs, USA) was employed for the laser intensity measurement and triggering the acoustic signal detection. The acoustic signal was measured by a contact transducer (V103RM 1-MHz, Olympus, Japan) attached to the cuvette wall using ultrasonic gel (US-A, 0–80 °C, Helling Nord Test US-A, Germany) and recorded using Tektronix DPO4104B oscilloscope (Tektronix Inc., USA). Peak-to-peak amplitudes of the photoacoustic responses were used to quantify MIL-101 NPs, using calibration curve (Supplementary Fig. 3). The MIL-101 NP degradation time represents time of 85% decrease of the photoacoustic amplitude from its maximum to minimum value.

### In vitro experiments

NCI-H1299 human non-small cell lung carcinoma and B16-F1 mice melanoma cell lines were cultured under standard conditions (37 °C, 5% $CO_2$) in DMEM medium supplemented with 10% FBS, penicillin/streptomycin and 2 mM L-glutamine.

To analyse the cytotoxicity of doxorubicin-loaded MIL-101 NPs, cells were seeded into 96-well plates at 3000 cells per well in full DMEM medium for 24 h. They were subsequently treated with free doxorubicin, unloaded MIL-101 NPs, and doxorubicin-loaded MIL-101 NPs at varying concentrations for 48 h. Then 20 μL of MTT solution (5 mg $mL^{-1}$) was added to each well and incubated for 2 h at 37 °C under 5% $CO_2$. After removing the MTT containing medium, 100 μL of DMSO was added to dissolve the MTT-formazan salt. The absorbance at 570 nm and 630 nm was measured using Infinite M1000 microplate reader (Tecan, Switzerland). The $IC_{50}$ value was calculated using DoseResp function of the OriginPro 9.1 software.

### Analysis of particle internalisation

NCI-H1299 cells were cultured in 12-well plates to 70% confluence under standard conditions (37 °C, 5% $CO_2$) prior to an addition of unloaded MIL-101 NPs of the concentration 25 μg $mL^{-1}$, and the cells were incubated for 15 min, 3 h, and 24 h. After the medium was removed, the cells were washed once with PBS and fixed in 5% paraformaldehyde for 15 min at room temperature. Next the iron core of the particles was stained by Perls Prussian blue for visualisation. Untreated cells served as the control. Cells were imaged using a DMI 6000B microscope (Leica Microsystems, Germany) equipped with a DMC2900 camera.

For the TEM analysis of particle internalisation, NCI-H1299 cells were cultivated in 100-$mm^2$ dishes to 70% confluence under standard conditions (37 °C, 5% $CO_2$) prior to an addition of the unloaded MIL-101 NPs. The particles were added at the concentration 25 μg $mL^{-1}$ and the cells were incubated for 15 min, 3 h, and 24 h. Next the cells were fixed in the 2.5% glutaric dialdehyde solution in 0.05 M cacodylate buffer (pH 7.4) augmented by 0.15-M sucrose for 2 h on ice, followed up by washing twice in cold cacodylate buffer augmented by 0.06 M sucrose

and incubated overnight at 4 °C. The post-fixation was carried out on the next day in 1% OsO₄ solution in cacodylate buffer augmented by 0.05 sucrose for 1 h on ice, followed up by washing 4 times for 5 min by cold milliQ water. Next the cells were carefully scrapped, transferred to tubes, dehydrated in a series of alcohol solutions of increasing alcoholic concentration, then in a series of araldite and acetone mixtures of increasing resin concentration and finally embedded in 1:1 mixture of araldite and EPON resins. Ultrathin sections of the specimen were produced using LKBIII ultramicrotome (LKB Bromma, Sweden) with glass knifes. Freshly cut sections were mounted on copper grids and stained with uranyl acetate and lead citrate. TEM images were obtained at a magnification of 4000× and 10,000× using TEM LIBRA 120 (Carl Zeiss, Germany) operating with WinTEM 1.3 software.

### In vivo experiments

The toxicity and biodistribution of the MIL-101 NPs in vivo were evaluated in BALB/c mice. The MIL-101 NPs were administered via the tail vein dosed 25 mg kg⁻¹. Next the mice were euthanised by cervical dislocation, and the liver, spleen, lungs, heart, and kidneys were collected for analysis.

For histological examination, the samples of internal organs were fixed in 4% formaldehyde for 24 h, dehydrated in a series of alcohol solutions of increasing alcoholic concentration, and embedded in paraffin. 4-µm thick paraffin sections were stained with haematoxylin & eosin, Van Gieson's or eosin & Perls Prussian blue. Microscopy studies were carried out using Leica DM4000 B LED light microscope (Leica Microsystems, Germany) equipped with a digital camera Leica DFC7000T.

In addition to the histopathological description, acquired images were analysed semi-quantitatively to score the results of the examination (Supplementary Tables 4, 6, 7). Morphological signs of the target organs in every slide were evaluated with 0-to-3 point system: 0−no sign; 1−the least pronounced sign; 2−moderately pronounced sign; 3−the most pronounced sign. The slides were blindly evaluated by a qualified pathologist. Morphometry of the Kupffer cells number in the liver was carried out in 10 fields of the microscope at 200× magnification for each animal using Leica Application Suite, version 4.9.0.

The nanoparticle biodistribution analysis was carried out using 50-100 mg tissue specimens from the respective organs, which were dissolved in a 3-fold volume of nitric acid. Next, the solution was heated to 60 °C for 30 min, diluted 10-times with distilled water and centrifuged at 1000 g for 15 min. Iron concentration was measured by ICP-MS (NexION 2000, Perkin Elmer, USA).

The diffusion of fluorescent dye from the dye-loaded MIL-101 NPs to the lung tissue was studied, as follows. The lungs were quickly excised in 15, 30, 60, and 180 min after the intravenous administration of rhodamine-123-loaded MIL-101 NPs. The lung tissue was flattened between two cover glasses and fixed in a holder. The microscopic analysis was carried out using a confocal laser scanning microscope Axio Observer Z1 LSM 710 DUO (Carl Zeiss, Germany) under the excitation at 514 nm and detection in the spectral range of 522–600 nm. The microscope was operated with Zen Black 3.0 software. Images were obtained by merging several fields of view. Fluorescence intensity of the tissue was determined by using ImageJ 1.8.0 software as the mean over the tissue away from the air alveolar borders and identified blood vessels (see example at Supplementary Fig. 15). Brightness from at least 9 lines were averaged for each data point.

### Tumour treatment

B16-F1 cells were cultivated in the DMEM medium supplemented with 10% FBS and 1 mM of L-glutamine under standard conditions. Upon reaching 100% confluency, the cells were trypsinised, collected, diluted in PBS and used for injection to animals. An experimental lung metastasis model was established by injection of 1×10⁵ B16-F1 cells in 100 µL of PBS into the tail vein of C57Bl/6 mice.

For the treatment of metastases at early stage, the animals were randomised and on Day 1, Day 3 and Day 5 after the tumour inoculation were treated with 200 µg of MIL-101 NPs loaded with 72.4 µg of doxorubicin, the same mass of free doxorubicin, or PBS. Then, on Day 10 and Day 15 after the tumour inoculation, mice were euthanised, lungs were collected and fixed with 4% formaldehyde for 24 h. For the treatment of metastases at late stage, treatment was performed at Days 7, 9 and 11, while the mice were euthanised at Day 18. The number of the lung surface tumour nodules was analysed using a stereomicroscope. Photographs of the lungs were captured by a DCM-510 SCOPE camera (Micromed, Russia). The size of the metastases was analysed using ImageJ 1.8.0 software, and all observed nodules in the images were included in the analysis. The survival analysis was carried out by monitoring animals every day after the tumour inoculation.

### Statistical analysis

All experiments on the MIL-101 NP degradation were performed at least in triplicates. All measurements were taken from distinct samples if not statement otherwise. Data are presented as mean ± standard deviation (SD). In Figs. 6,7 boxplots represent median values, interquartile ranges and Tukey whiskers with individual data points superimposed. Normal distribution of quantitative data from the pathomorphological analysis were checked by Shapiro-Wilk normality test. Significant differences were determined using one-way ANOVA with Tukey post-hoc test. Significant differences of the survival curves were evaluated using two-sided log-rank test.

### Reporting summary

Further information on experimental design is available in the Nature Research Reporting Summary linked to this Article.

## Data availability

The statistical data for Figs. 1c−e, 2c, d, 2f, g, 3a, 3c−h, 4a−c, 5b and 5d, 6b, 6d, e, 7d−e are available in Source data file. Results of morphological analysis of histology and data used for numerical calculations are provided within the Supplementary Information. The statistical data for Supplementary Figs. 1c, d, 2c, d, 3, 4a, b, 5, 6, 7, 8, 9 are available in Source data file. Source data are provided with this paper.

## Code availability

The code used in the theoretical modelling described in Supplementary Note 1 is available at https://github.com/zelepukiny/FlaRE-delivery[49].

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

## Acknowledgements

The work was supported by the Ministry of Science and Higher Education of the Russian Federation, agreement no. 075-15-2020-773 awarded

to S.M.D. We thank Prof. A.B. Schekhter (Sechenov First Moscow State Medical University) for the help with the histology evaluation and A.S. Sogomonyan (Shemyakin-Ovchinnikov Institute of Bioorganic Chemistry RAS) for the help with the cytotoxicity evaluation.

## Author contributions

I.V.Z. designed experiments. A.V.I. and A.V.Z. designed and performed theoretical modelling. I.V.Z. and O.Yu.G. synthesised and characterised nanoparticles and their degradation. K.G.S. designed and performed in vitro experiments. K.G.S. and E.V.B. performed TEM experiments. N.B.S. performed histology analysis. I.V.Z., O.Yu.G. and A.B.V. performed in vivo experiments. S.M.D. and A.V.Z. supervised the study. I.V.Z., K.G.S., A.V.Z. and S.M.D. analysed data, wrote, and corrected manuscript.

## Competing interests

The authors declare no competing interests.
