## [Peer Review File · Nature Communications]

Flash drug release from nanoparticles accumulated in the targeted blood vessels facilitates the tumour treatmentREVIEWER COMMENTS

Reviewer #1 (Remarks to the Author):

The work of Zlepukin et al. is well done and written but the in vivo experiment raises concerns. The mouse experiment is not a tumor treatment model but rather a chase study in which shortly after injection of tumor cells, which will get stuck in finer vessels, nanosystems are injected. These NPs will get stuck in the same regions. The systemic approach is therefore not realistic and tuned towards a response by the tumor cell clusters.

The nanosystems used are short term sustained release systems, which will have systemic effect, i.e. toxicity, and will not per se in a metastasis model deliver preferentially to the tumor.

Reviewer #2 (Remarks to the Author):

The presented manuscript is of high importance to the cancer- and nanotechnology-related communities, especially for research oncologists working with melanoma. The authors presented comprehensive convincing theoretical and experimental data. However, some details and relevant discussions are missed.

1. The authors use a word "new" or "novel" many times (e.g., two times in an Abstract) sometimes without clear justification. For example, in abstract: "We introduce a new anticancer drug delivery concept termed FlaRE (Flash Release in Endothelium), which relies on rapid accumulation of drug-loaded nanoparticles in the tumour-surrounding blood vessels...". There is no quite clear an experimental confirmation in animal model in vivo that nanoparticles indeed selectively accumulates in tumour-surrounded vessels. What is a novel mechanism of accumulation compared to known EPR, capturing relatively large nanoparticles in small vessels, or molecular targeting?

2. According to Figure 5, the nanoparticles can penetrate into tissue surrounding the vessels. What is their possible role in tumor killing?

3. Some laser parameters are missed, e.g., a pulse duration or energy fluence. Many nanoparticles have a tendency to aggregations in different environment during short time. During photoacoustic monitoring a laser can destroy them even at low energy leading to the signal degradation. If it is possible scenario case, it must be discussed.

4. Is there clear comparison of the theoretical and experimental data (e.g., curves)?

5. Supplementary Figure 3: why the photoacoustic signal demonstrates a nonlinear behavior with a concentration increasing from 0.008 to 0.2 g/L?

6. There is no discussion of different cancer lines selection for in vitro and in vivo experiments.

7. It is not clear why drug alone at relatively high concentration has no effect (Fig.6b).

8. Different terminologies and abbreviations were used for description either the same agents (e.g., particles, nanoparticles, or MIL-11 NPs) or phenomenon (e.g., photoacoustic, acoustic, or optoacoustic [Figs.2g, 4a]).

9. A black curve in Figure 2d is missed.

To Reviewer #1:

Dear Reviewer #1,

First of all, we would like to sincerely thank you for a scrupulous evaluation of our manuscript and for the valuable comments. Please find below the detailed answers to the questions mentioned in your review (the Reviewer's remarks are presented in **bold**, and our answers follow in plain text, quotations from the manuscript are shown in *italic*).

The work of Zlepukin et al. is well done and written but the in vivo experiment raises concerns. The mouse experiment is not a tumor treatment model but rather a chase study in which shortly after injection of tumor cells, which will get stuck in finer vessels, nanosystems are injected. These NPs will get stuck in the same regions. The systemic approach is therefor not realistic and tuned towards a response by the tumor cell clusters.

Indeed, in the previous version of manuscript the treatment was performed on Day 1 after administration of cancer cells. We agree with the Reviewer that this scheme modelled rather "metastases prevention" than their treatment. To address this issue the related part of the manuscript has been rewritten.

Additionally, we performed new experiments to confirm the feasibility of our approach for metastasis treatment (Figure 7). To do that we used the previously described model of late-stage B16-F1 metastases [1,2]. In this case the therapy with doxorubicin-loaded MIL-101 NPs started on Day 7 after administration of cancer cells. At this stage, the presence of tumour nodules and their extravasation from the lung capillaries to surrounding tissue was clearly observed at histological images (Figure 7b and Supplementary Figure 13). The size of the tumour nodules was $95 \pm 40 \mu\text{m}$ (mean \pm SD).

Supplementary Fig. 13. Representative histological images, showing extravasation of B16-F1 melanoma (blue arrows) from blood vessel to surrounding tissue. Scale bars = 100 μm .

Doxorubicin-loaded MIL-101 NPs showed better efficiency for late-stage melanoma treatment as compared to free doxorubicin. We observed 4.3-fold and 1.5-fold decrease in melanoma nodule quantity for groups treated with doxorubicin-loaded MIL-101 NPs and doxorubicin, respectively, when compared with the untreated control (Figure 7d). The new results were described in the text:

For the late-stage metastasis model,³⁹ 1×10^5 of B16-F1 cells were administrated via the tail vein (Fig. 7a). Histological evaluation of the lungs harvested from 3 random mice on Day 7 demonstrated presence of numerous micro-metastases of $95 \pm 40 \mu\text{m}$ size. The tumour cells started extravasation from the capillaries to the surrounding lung tissue (see Fig. 7b and Supplementary Fig. 13). Doxorubicin-loaded MIL-101 NPs were systemically administered on Days 7, 9, and 11 via the tail vein in a dosage of 10 mg kg^{-1} . Control groups were treated with free doxorubicin in the equivalent loaded dose or PBS in the same regimen, respectively. The animals were sacrificed on Day 18, their lungs were collected, metastases were counted and sized under a dissecting microscope.

In general, doxorubicin-loaded MIL-101 NPs were less effective for a late-stage metastases treatment than for early-stage one. Nevertheless, the MIL-101 NPs still show better therapeutic

efficacy than free doxorubicin. The number of melanoma nodules in groups treated with doxorubicin-loaded MIL-101 NPs and free doxorubicin decreased 4.3-fold and 1.5-fold, respectively, compared to the PBS-treated negative control. The median size of the nodules decreased from 482 μm in PBS-treated group to 317 μm for the doxorubicin-loaded MIL-101 NP treated group and to 316 μm for the free doxorubicin treated group. We suggest that for the long-term metastasis model MIL-101 NPs retain the better treating efficiency compared to a free drug, which supports the FlaRE drug delivery model.

Fig. 7. Doxorubicin-loaded MIL-101 NPs effectively treat melanoma in late-stage pulmonary metastasis model. *a* Treatment schedule. *b* Representative histological images, showing extravasation of B16-F1 melanoma cells from blood vessel to surrounding tissue. Scale bar = 100 μm . *c* Representative images of the lungs, collected on Day 18 post cancer cell administration. Mice were treated with PBS (top line), doxorubicin (middle line), or doxorubicin-loaded MIL-101 NPs (bottom line). Blue arrows point to melanoma metastases. $n = 3$ lungs. *d* Quantification of the surface nodule metastases in the harvested lungs. *e* Analysis of the nodule size in the harvested lungs. Tukey boxplot

was used for data visualisation. In **d,e** $n = 8$ animals for each group. ANOVA with Tukey post-hoc test was used for statistical analysis.

[1] - DOI: 10.1126/sciadv.aax9250

[2] - DOI: 10.1016/j.ijpharm.2020.119446

The nanosystems used are short term sustained release systems, which will have systemic effect, i.e. toxicity, and will not per se in a metastasis model deliver preferentially to the tumor.

The reviewer's comment is valid. We employed the short-term sustained release nanosystems, e.g. doxorubicin-loaded MIL-101 NPs, to exert systemic effect on the entire metastasised organ, the lungs. In the demonstrated case, the MIL-101 NPs predominantly (~60% ID) accumulated in the lung vasculature and rapidly released loaded doxorubicin. The gradient-driven drug permeated to the lung parenchyma, killed most cancer cells with little toxicity to normal cells, as revealed by the lung tissue histology analysis (no pathological changes for the selected therapeutic dosage, Supplementary Note 3). Besides, the metastases appeared to develop in the perivascular regions as clearly seen in Figure 7b – as such, the tumour nodes can be deemed vascularised.

We wish to stress that our approach based on short-term sustained release nanosystems is suitable for treatment of both metastatic and vascularised solid tumours. This approach was demonstrated experimentally for metastases and generalised theoretically for primary tumours. The proposed approach can be beneficial for solid tumours, where passive particle accumulation in microcapillaries has been demonstrated. Several related studies were described in the discussion.

This work represents successful accomplishment of an important milestone: drug delivery to targeted vascularised organs and treatment of vascular-surrounding metastases. We hope that the reviewer appreciates the scale, the level of achievements, and the promise of this work.

We added the following text to Discussion:

Suggested approach of FlaRE delivery differs from the EPR-based drug delivery strategies, which currently dominate nanomedicine. This approach is based on a rapid degradation of drug-loaded nanocarriers accumulated in the targeted vasculature and doesn't require extravasation of nanoparticles for effective drug delivery. In this study, the efficacy of FlaRE approach was experimentally demonstrated in treatment of metastatic tumours in lungs. MIL-101 NPs were preferentially trapped in lung vasculature (~60% ID), likely due to their increase in size and to the change in their ζ -potential to the positive values after the phosphate binding in the bloodstream. Similar approach for targeting of an entire diseased organ has been demonstrated previously for selective organ targeting (SORT) nanoparticles and for RBC-

hitchhiking technology.^{42,43} Combination of rapidly degradable nanoparticles with specific targeting can further increase efficacy of FlaRE approach. The existing examples include active targeting of the inflamed vessel endothelium in lungs and brain due to NP anchoring to ICAM-1 and PECAM-1 receptors.⁴⁴

The proposed approach might be theoretically generalised for treatment of primary tumours, if the passive accumulation of particles in tumour microcapillaries may be demonstrated. The reduced blood speed in the tumour vessels was one of the key mechanisms of the particle adhesion to the endothelium reported by Decuzzi, et al (5% ID g⁻¹)⁴⁵ and Parakhonskiy, et al (30% ID g⁻¹).⁴⁶ Active endothelial targeting of the solid tumour vessels has been also reported.⁴⁷ The accelerated angiogenesis provides several other targeting options – anti-fibronectin EDB domain antibody fragment L19 as well as many RGD and NGR peptides binding integrins to the angiogenic endothelium may be employed.⁴⁸

To Reviewer #2:

Dear Reviewer #2,

First of all, we would like to sincerely thank you for a scrupulous evaluation of our manuscript and for the valuable comments. Please find below the detailed answers to the questions mentioned in your review (the Reviewer's remarks are presented in **bold**, and our answers follow in plain text, quotations from the manuscript are shown in *italic*).

The presented manuscript is of high importance to the cancer- and nanotechnology-related communities, especially for research oncologists working with melanoma. The authors presented comprehensive convincing theoretical and experimental data. However, some details and relevant discussions are missed.

1. The authors use a word “new” or “novel” many times (e.g., two times in an Abstract) sometimes without clear justification. For example, in abstract: “We introduce a new anticancer drug delivery concept termed FlaRE (Flash Release in Endothelium), which relies on rapid accumulation of drug-loaded nanoparticles in the tumour-surrounding blood vessels...”.

We have carefully checked the manuscript for any unnecessary use of the terms “new” and “novel” in relation to the developed drug delivery method.

There is no quite clear an experimental confirmation in animal model in vivo that nanoparticles indeed selectively accumulates in tumour-surrounded vessels.

We agree with the Reviewer that there was no selective accumulation of nanoparticles exclusively in the tumour-surrounding vessels. In our study we demonstrated preferential, but non-specific accumulation of MIL-101 NPs in lung vessels (Figure 5a). In addition, we show that B16-F1 metastases proliferate near the vessels and extravasates to surrounding tissues in 1 week (Supplementary Figure 13). To avoid the confusion, we clarified the text, highlighted by the Reviewer, as well as the other relevant texts. For example:

“This approach relies on enhanced drug-loaded nanocarrier accumulation in vessels of the target tumour or metastasised organ...”

Supplementary Fig. 13. Representative histological images, showing extravasation of B16-F1 melanoma (blue arrows) from blood vessel to surrounding tissue. Scale bars = 100 μ m.

What is a novel mechanism of accumulation compared to known EPR, capturing relatively large nanoparticles in small vessels, or molecular targeting?

The following steppingstones are pivotal for the FlaRE concept realisation:

1. Enhanced accumulation of the particles in the vessels of a target organ or tumour-surrounding vessels.
2. Rapid release of a drug encapsulated in the nanoparticles.

We stress that unlike in the EPR effect, no extravasation of drug-loaded nanocarriers is required. It is not critical how the enhanced intracapillary accumulation of nanocarriers in the

tumour vessels is achieved. In this work, the enhanced accumulation was achieved by trapping nanoparticles in the lung microcapillaries, while other methods have been reported in the literature. We addressed these studies in Discussion:

Suggested approach of FlaRE delivery differs from the EPR-based drug delivery strategies, which currently dominate nanomedicine. This approach is based on a rapid degradation of drug-loaded nanocarriers accumulated in the targeted vasculature and doesn't require extravasation of nanoparticles for effective drug delivery. In this study, the efficacy of FlaRE approach was experimentally demonstrated in treatment of metastatic tumours in lungs. MIL-101 NPs were preferentially trapped in lung vasculature (~60% ID), likely due to their increase in size and to the change in their ζ -potential to the positive values after the phosphate binding in the bloodstream. Similar approach for targeting of an entire diseased organ has been demonstrated previously for selective organ targeting (SORT) nanoparticles and for RBC-hitchhiking technology.^{42,43} Combination of rapidly degradable nanoparticles with specific targeting can further increase efficacy of FlaRE approach. The existing examples include active targeting of the inflamed vessel endothelium in lungs and brain due to NP anchoring to ICAM-1 and PECAM-1 receptors.⁴⁴

The proposed approach might be theoretically generalised for treatment of primary tumours, if the passive accumulation of particles in tumour microcapillaries may be demonstrated. The reduced blood speed in the tumour vessels was one of the key mechanisms of the particle adhesion to the endothelium reported by Decuzzi, et al (5% ID g⁻¹)⁴⁵ and Parakhonskiy, et al (30% ID g⁻¹).⁴⁶ Active endothelial targeting of the solid tumour vessels has been also reported.⁴⁷ The accelerated angiogenesis provides several other targeting options – anti-fibronectin EDB domain antibody fragment L19 as well as many RGD and NGR peptides binding integrins to the angiogenic endothelium may be employed.⁴⁸

Moreover, since the requirement of the extravasation (essential for EPR-based drug delivery strategies) is abrogated in favour of maximum delivery efficiency to the vasculature of the target tissue, the dimensional range of optimal drug carriers may be expanded towards sub-micron sizes. This provides a new view on nanomedicine design. For example, sub-micrometre discoidal particles have been reported to be effective as intravascular carriers to maximise accumulation in the target organ.^{45,47}

2. According to Figure 5, the nanoparticles can penetrate into tissue surrounding the vessels. What is their possible role in tumor killing?

Indeed, we observed accumulation of nanoparticles in the lung tissue as shown in Figure 5. Although the MIL-101 NPs could not be recognised as therapeutic agent, the terephthalic acid released upon nanoparticles degradation might have some effect on the tumour cells.

To prove that unloaded MIL-101 NPs have limited effect on cell viability we performed additional experiments and evaluated their cytotoxicity in B16-F1 melanoma cells using MTT assay. Supplementary Figure 9 shows, that MIL-101 NPs itself have not significantly affected cell viability even in concentration as high as 125 mg/L. We did not observed signs of their toxicity *in vivo* in the sites of their predominant accumulation (Supplementary Note 3).

These new results were described in the text:

These observations were confirmed by the results of cytotoxicity assay in B16-F1 cells. The viability was higher than 70% even in MIL-101 NP concentration as high as 125 mg L⁻¹. The IC₅₀ value for doxorubicin-loaded MIL-101 NPs was 1.02 ± 0.28 µg mL⁻¹, which was equivalent to that of free doxorubicin with IC₅₀ value of 418 ± 106 ng mL⁻¹ (Supplementary Fig. 9). The difference in relative particle toxicity for the cell lines may be attributed to variations in their metabolism, repertoire of cell surface receptors, and subsequent changes in cell-nanoparticle interactions.

Supplementary Fig. 9. MTT cytotoxicity assay of MIL-101 NPs, doxorubicin-loaded MIL-101 NPs and free doxorubicin with B16-F1 melanoma cells. The cell viability is presented as % ratio normalised to the non-treated control cells. *n* = 6 samples analysed per data point.

3. Some laser parameters are missed, e.g., a pulse duration or energy fluence.

The highlighted laser parameters were added to the text of the manuscript:

MIL-101 NP aqueous colloids were illuminated with a nanosecond pulsed Nd:YAG Quantas Q1B laser (Quantum Light Instruments, Lithuania). The 2nd harmonic at 532 nm was employed with pulse repetition rate of 20 Hz, pulse duration of 5 ns and energy per pulse of 5 mJ. The laser beam was expanded to fill a 1-cm cuvette width by a diverging lens, to minimize nanoparticle damage by laser irradiation.

Many nanoparticles have a tendency to aggregations in different environment during short time. During photoacoustic monitoring a laser can destroy them even at low energy leading to the signal degradation. If it is possible scenario case, it must be discussed.

The following discussion and additional experiments have been added to the text of the manuscript:

Laser irradiation during the photoacoustic measurement can thermally or mechanically damage the nanoparticles enhancing their degradation.²⁵ Moreover, particle sedimentation during the measurement may lead to the underestimation of their quantity in environment. To reduce influence of these factors on the measurement results, we expanded the laser beam over the width of a 1-cm cuvette, at the same time lowering laser energy and enhancing irradiation volume. Supplementary Fig. 4 shows that the optical properties and photoacoustic responses showed no changes after 15-min laser exposure in for intact MIL-101 NPs in water and for degraded MIL-101 NPs in PBS. It supports our notion that sedimentation and photodegradation had minimal influence on the kinetic analysis during the measurement period.

Supplementary Fig. 4. *a* extinction spectra and *b* photoacoustic signals of MIL-101 NPs in water and their degraded forms in PBS, before and after 15-min laser irradiation by photoacoustic setup.

4. Is there clear comparison of the theoretical and experimental data (e.g., curves)?

In the manuscript we fitted the experimental data of drug accumulation in lung tissue *ex vivo* measured by the Rhodamine 123 fluorescence intensity (model drug) to our theoretical model. This data was described in the text:

We wish to point out that we performed validation of our theoretical model by its comparison with the experimental data of the fluorescent dye (Rhodamine 123) biodistribution in the lung tissue *ex vivo*. The fitting results are presented in Figure 5d. These results were described in the text:

The formulated dye-loaded MIL-101 NPs were intravenously administered at a dose of 25 mg kg⁻¹. The mice were sacrificed at time points ranging from 15 min to 3 h post injection and the lungs were collected for investigation. The dye distribution in unprocessed interstitial tissue was immediately imaged by laser-scanning fluorescent confocal microscopy. We observed that the fluorescence signal of tissue gradually increased for the first 30 min after the injection, followed by an exponential decrease over a 3-h period. The acquired data were compared with our theoretical model, using an algorithm described in Supplementary Note 2. In brief, we fitted the kinetics of the bound drug concentration, obtained in our theoretical model (Fig. 1c), to an analytical function, where the distance from a capillary was used as a fitting parameter. This function well fitted the experimental data of the dye distribution in lungs (Fig. 5d), providing further validation of the FlaRE drug delivery model.

Fig. 5. ... d Kinetics of the mean fluorescence intensity of the lung tissue. Data were fitted using the developed theoretical model (red line). Grey line marks the tissue autofluorescence level. $n = 3$ mice.

Supplementary Note 2. Theoretical fitting of rhodamine 123 kinetics in the lung tissue

The kinetics of the bound drug concentration at a fixed distance from a capillary obtained in our theoretical model (c.f. Supplementary Fig. 1 sampled at the fixed distance at several time points), was fitted to an analytical function of the drug concentration, C , where the distance from a capillary, r was set to $45 \mu\text{m}$. The analytical function for the rhodamine 123 kinetics in the lung tissue was constructed as derived by J. Crank⁶ for an infinite source line in a homogeneous medium characterised by the substance desorption rate $\phi(t) = \frac{\Phi'}{l_c} [s^{-1} \text{mm}^{-1}]$:

$$C = \frac{1}{4\pi D} \int_0^t \phi(t') \exp\left[-\frac{r^2}{4D(t-t')}\right] \frac{dt'}{t-t'}$$

where D – diffusion constant, t - time.

The numerical solution of this equation was used to fit the experimental data by the least square method using Mathematica 13.1 software. This function was used to fit the experimental data of the dye distribution. The results of the modelling are presented in Fig. 5d.

Supplementary References

6. J. Crank, *The Mathematics Of Diffusion* (Clarendon Press, Oxford, 1975).

5. Supplementary Figure 3: why the photoacoustic signal demonstrates a nonlinear behavior with a concentration increasing from 0.008 to 0.2 g/L?

The obtained photoacoustic data was approximated via a linear function ($y = 0.013 + 0.0761x$, red line, Supplementary Figure 3). For clarity, we provided the following Additional Figure 1, where the linearity is shown on a linear scale:

Additional Figure 1. Calibration plot of the photoacoustic system for quantification of the concentration of MIL-101 NPs. $n = 3$ samples. Red line shows linear fitting of the data points. Blue line shows photoacoustic signal from water $\pm 3\sigma$ electronic noise level.

Nevertheless, log-log scale was chosen to visualise the data in the broad concentration range. The line function shows good linear fitting of the experimental data in 10^{-3} g/L to 0.125 g/L concentration range, with minor deviations attributed to measurement errors. The increased laser light attenuation along the propagation path in the cuvette led to an underestimation of the MIL-101 NP concentration in the range exceeding 0.125 g/L.

Supplementary Fig. 3. Calibration plot of the photoacoustic system for quantification of the concentration of MIL-101 NPs. $n = 3$ samples. Blue line shows photoacoustic signal from water ± 3 -fold electronic noise level. Red line shows linear fitting of the data points. Linear range was $10^{-3} - 0.125 \text{ g L}^{-1}$.

6. There is no discussion of different cancer lines selection for in vitro and in vivo experiments.

To address Reviewers comment, first, we performed toxicity evaluation of MIL-101 NPs and doxorubicin-loaded MIL-101 NPs *in vitro* using B16-F1 cell line, which we use *in vivo* (see answer on the 2nd question). In addition, following short discussion was added to the text:

We hypothesised the cells would uptake released doxorubicin from the incubation media rather than from the endocytosed particles. To test this hypothesis, we performed MTT cytotoxicity assay in NCI-H1299 human non-small cell lung carcinoma and B16-F1 murine melanoma cell lines. The choice of NCI-H1299 cells was justified by their universal use for in vitro studies of lung cancer in humans.³⁵ The B16-F1 cells have been successfully applied for establishing a murine model of extremely aggressive lung metastases.³⁶

7. It is not clear why drug alone at relatively high concentration has no effect (Fig.6b).

We added discussion of this fact in the manuscript:

*Interestingly, while free doxorubicin in 3.6 mg kg^{-1} concentration reduced the quantity of metastases (Fig. 6d), it did not significantly affect the mice survival time (Fig. 6b). Our observation correlates with the previous studies which demonstrated no effect of high doses of doxorubicin on prolongation of life duration of mice.^{39,40} For example, administration of free doxorubicin in 5.4 mg kg^{-1} dose, which is 1.5 times higher than in present study, increased median survival time in murine melanoma model from 29 to 32 days only.³⁹ We assume that this effect may be explained by the low influence of free doxorubicin on proliferation rate of melanoma cells *in vivo*. On Day 15 after cell administration, we have not observed any decrease in the mean size of metastases in the free doxorubicin-treated group (Fig. 6e). I.J. Fidler have shown that only a small number of initial micrometastases evolve to a life-threatening macrometastases stage.⁴¹ In case of present study this notion may be interpreted as follows. The administration of free doxorubicin may reduce the number of smaller metastases on their initial development stage; however, the rapid growth of the remaining ones would eventually lead to a death outcome in a short time. Encapsulation of the drug into MIL-101 NPs increases doxorubicin delivery efficacy, which manifests both in the decrease of metastasis size on Day 15 and in the prolongation of mice survival. It validates the proposed*

FlaRE delivery concept for treatment of disseminating melanoma at the early stages of metastases development.

8. Different terminologies and abbreviations were used for description either the same agents (e.g., particles, nanoparticles, or MIL-11 NPs) or phenomenon (e.g., photoacoustic, acoustic, or optoacoustic [Figs.2g, 4a]).

We have carefully checked the consistency of all the terms and abbreviations used in the manuscript and figures. Particularly, MIL-101 (Fe) metal-organic frameworks have been termed MIL-101 NPs. The photoacoustics related wording has also been brought to single terminology.

9. A black curve in Figure 2d is missed.

Thank you for this comment. Black curve was presented in the graph, but it was overlapped with blue curve. We redraw this Figure in the manuscript:

Fig.2 ... d Extinction spectra of MIL-101 NPs 24-h post incubation in ethanol (green), water (blue), and PBS buffer (red). Inset – images of cuvettes with NP solutions in dispersants, $n = 1$ measurement.

REVIEWERS' COMMENTS

Reviewer #1 (Remarks to the Author):

Authors answered the questions and added appropriate data.

Reviewer #2 (Remarks to the Author):

I am satisfied with the author's responses to my questions.
A manuscript can be published.